# Mesmerize is a dynamically adaptable user-friendly analysis platform for 2D and 3D calcium imaging data

Kushal Kolar [1✉], Daniel Dondorp[1], Jordi Cornelis Zwiggelaar [1,2], Jørgen Høyer [1,2] & Marios Chatzigeorgiou [1✉]

Calcium imaging is an increasingly valuable technique for understanding neural circuits, neuroethology, and cellular mechanisms. The analysis of calcium imaging data presents challenges in image processing, data organization, analysis, and accessibility. Tools have been created to address these problems independently, however a comprehensive user-friendly package does not exist. Here we present Mesmerize, an efficient, expandable and user-friendly analysis platform, which uses a Findable, Accessible, Interoperable and Reproducible (FAIR) system to encapsulate the entire analysis process, from raw data to interactive visualizations for publication. Mesmerize provides a user-friendly graphical interface to state-of-the-art analysis methods for signal extraction & downstream analysis. We demonstrate the broad scientific scope of Mesmerize's applications by analyzing neuronal datasets from mouse and a volumetric zebrafish dataset. We also applied contemporary time-series analysis techniques to analyze a novel dataset comprising neuronal, epidermal, and migratory mesenchymal cells of the protochordate *Ciona intestinalis*.

[1] Sars International Centre for Marine Molecular Biology, University of Bergen, 5006 Bergen, Norway. [2]These authors contributed equally: Jordi Cornelis Zwiggelaar, Jørgen Høyer. ✉email: kushalkolar@gmail.com; Marios.Chatzigeorgiou@uib.no

Large-scale calcium imaging of neuronal activity in populated brain regions, or entire animals, has become an indispensable technique in neuroscience research. The analysis of calcium imaging datasets presents significant challenges in the domains of image preprocessing, signal extraction, dataset organization, downstream analysis, and visualizations. As a result, the analysis of calcium imaging data requires computational expertise that are rather uncustomary among biologists. Numerous state-of-the-art packages, such as the Caiman library[1], Suite2p[2],SIMA[3], EZCalcium[4] and ImageJ[5], provide users with a myriad of options for image preprocessing and ROI/signal extraction. Workflow management tools for neurophysiological analysis, such as DataJoint[6] and NWB[7], provide programmers with tools for dataset organization. Users with computational training often incorporate these tools using custom-written scripts or spreadsheets. In contrast, biomedical scientists with little or no programming experience would immensely benefit from a user-friendly platform to organize, analyze, visualize, and share 2D and 3D calcium imaging data.

An important attribute of such a platform would be the ability to seamlessly incorporate cutting-edge tools that will readily address current and future technical challenges. The immense growth we have seen over the last decade in new imaging technologies combined with the ever-increasing palette of genetically encoded indicators have fueled an increase in the temporal and spatial resolution of the acquired datasets. Calcium imaging is not only a workhorse technique for monitoring brain-wide activity, but it is becoming increasingly popular in the dissection of developmental and physiological processes at the level of entire embryos or organs. These types of information-rich datasets are characterized by the presence of large populations of morphologically and functionally diverse, tightly packed, cells that exhibit diverse activity profiles, making downstream processing challenging. In particular, the analysis of 2D and 3D calcium imaging datasets poses significant technical hurdles across multiple domains including those of image preprocessing, signal extraction, dataset organization, downstream analysis, and visualization.

One of the greatest challenges that modern biomedical research faces is compliance with FAIR data (Findable, Accessible, Interoperable, and Reusable) principles, which aim to set new and robust standards in terms of reproducibility and data sharing. However, even some of the most advanced analysis pipelines rely on custom-written scripts and spreadsheets, without a standardized system to organize and functionally link raw imaging data, analysis procedures, and visualizations[8,9]. This greatly impedes the reproducibility of the work even when the raw data are available[8–10]. State-of-the-art project management tools, such as OMERO[11], Biaflows[12], Cytomine[13], OpenBIS[14], and KNIME[15], are geared towards cell biology and histological analysis, and are not suited for neurophysiological or calcium imaging analysis (Table 1). Most crucially, none of these tools support the rich and comprehensive annotations necessary for most experiments in the field of neuroscience. For example, the analysis of neurophysiological experiments often requires temporal mapping of complex combinations of stimuli and behavioral annotations that directly correspond to the imaging data (Table 1). There are also experimental scenarios where the cells or regions of interest (ROIs) require a combination of annotation tags (text/numerical labels) describing features such as the cell type, morphology, or identity, which can be mapped back to the corresponding cell(s) or ROI(s). Finally, for publication, authors have to produce figures integrating all of the above (i.e. the calcium imaging data, the annotations, and the downstream analysis) to effectively and coherently convey the biological findings. While there are many tools for producing basic static visualizations, there is an urgent need for a software platform that can produce interactive visualizations where the imaging data and analysis history of every datapoint can be instantly retrieved[8,9,16]. Interactive and traceable visualizations have various applications, such as quality control[8], reproducibility[9,16,17], and allowing for a better understanding of experiments and the underlying biology[8].

From the examination of the tools currently available for calcium imaging analysis and bio-imaging project management (Table 1), we demonstrate that there is currently no tool that provides a comprehensive suite of features necessary for calcium imaging analysis and project management, i.e. image processing, ROI extraction, project organization, downstream analysis, and interactive visualizations. To address these challenges, we created Mesmerize—a free and open-source comprehensive platform that encapsulates these requirements within a reproducible system. The Mesmerize platform also provides graphical user interfaces (GUI) for the analysis and visualization of 2D and 3D datasets, thereby allowing biomedical scientists to create FAIR (Findable, Accessible, Interoperable, and Reusable) datasets[10,18] within a flexible system that can be adopted by a wide variety of researchers who work on diverse biological problems. Mesmerize is not a pipeline, but rather a highly modular platform that presents users with many options along each step of their specific user-defined calcium imaging analysis workflow. Consequently, this flexible design allows developers to easily add new or customized modules for image processing, analysis, and visualization. In summary, the ability to create modular and adaptable workflows grants Mesmerize a very broad scope of applicability across a variety of labs in various fields of neuroscience. For example, it may be used to study whole-brain dynamics, sensory-motor integration systems, or activity defects in disease models. Beyond neuroscience, Mesmerize has the potential to be transformative in the hands of developmental biologists and physiologists interested in mapping embryonic and post-embryonic calcium dynamics of specific tissues/organs or entire embryos. Mesmerize lets users create and dynamically curate an unlimited number of categorical labels that map to entire imaging sessions, single ROIs, and temporal periods. This rich and complex annotation capability goes beyond standard neurobiological annotations such as behavioral correlates or sensory stimuli and can be extended to developmental stages, shared gene expression patterns, morphological and phenotypic cell-type descriptors, and subcellular compartments to a name a few. This flexibility means that Mesmerize is broadly suitable for cell biologists, developmental biologists, and other specialties beyond neuroscience. In scenarios where the analysis workflows require further tailoring, Mesmerize can serve as a blueprint for future platforms that seek to encapsulate data analysis, project organization, and interactive traceable visualizations in other fields.

As introduced above, calcium imaging analysis usually requires the following components: (1) preprocessing and ROI/signal extraction; (2) data annotation and organization; (3) downstream analysis; and (4) visualization. Mesmerize provides end-users with extensive graphical interfaces for each of these components to analyze their 2D and 3D datasets. Users with basic Python or scripting skills can utilize the API to implement more customized or complex analysis. We have built the graphical interfaces using the Qt framework due to its maturity and extensive developer community. All data structures are well-documented and built using pandas DataFrames[19] and numpy arrays[20,21], both highly prevalent and mature libraries. These features make Mesmerize a highly accessible platform, allowing users to easily integrate Mesmerize into their analysis workflows, or develop new customized modules.

**Table 1 Overview of various image analysis tools.**

| Package | Type | Suited for Ca imaging | 3D calcium imaging | Motion correction | ROI extraction | Project management | ROI annotation | Temporal annotation | Sample annotation | Graphical interfaces | Scripting interfaces | Downstream analysis | FAIR dataset creation | Visualization | Interactive visualization |
|---|---|---|---|---|---|---|---|---|---|---|---|---|---|---|---|
| Mesmerize | Platform | ✓ | ✓ | ✓ | ✓ | ✓ | ✓ | ✓ | ✓ | ✓ | ✓ | ✓ | ✓ | ✓ | ✓ |
| Caiman | Pipeline | ✓ | ✓ | ✓ | ✓ | ✗ | ✗ | ✗ | ✗ | L | ✓ | ✗ | ✗ | ✗ | ✗ |
| Suite2p | Pipeline | ✓ | ✓ | ✓ | ✓ | ✗ | ✗ | ✗ | ✗ | L | ✓ | L | ✗ | L | ✗ |
| EZCalcium | Pipeline | ✓ | ✗ | ✓ | ✓ | ✗ | ✗ | ✗ | ✗ | L | ✗ | ✗ | ✗ | ✗ | ✗ |
| SIMA | Pipeline | ✓ | ✓ | ✓ | ✓ | ✗ | ✗ | ✗ | ✗ | L | L | ✗ | ✗ | L | ✗ |
| S. A Romano | Pipeline | ✓ | ✓ | ✓ | ✓ | ✗ | ✗ | ✗ | ✗ | L | L | L | ✗ | L | ✗ |
| SamuROI | GUI Tool | ✓ | L | ✓ | ✓ | ✗ | L | ✗ | ✗ | L | ✗ | ✗ | ✗ | ✗ | ✓ |
| DataJoint | Workflow management | ✓ | ✓ | ✗ | ✗ | ✓ | ✗ | ✓ | ✓ | L | ✓ | ✓ | ✓ | ✗ | ✗ |
| OMERO | Platform | ✗ | ✗ | ✗ | ✓ | ✗ | L | ✗ | L | ✓ | ✓ | ✓ | ✓ | ✓ | ✗ |
| Biaflows | Platform | ✗ | ✗ | ✗ | ✓ | ✓ | ✗ | ✗ | L | ✓ | ✓ | L | ✓ | ✓ | ✓ |
| Cytomine | Platform | ✗ | ✗ | ✗ | ✓ | ✓ | ✓ | ✓ | ✓ | ✓ | ✓ | ✓ | ✓ | ✓ | ✓ |
| openBIS | Platform | ✗ | ✗ | ✗ | ✓ | ✓ | ✓ | ✓ | ✓ | ✗ | ✓ | ✓ | ✓ | ✓ | ✓ |
| KNIME | Platform | ✗ | ✗ | ✗ | ✓ | ✓ | ✓ | ✓ | ✓ | ✓ | ✓ | ✓ | ✓ | ✓ | ✗ |

An overview of various tools for calcium imaging analysis and dataset organization. The availability of various features for calcium imaging analysis, data annotation, data management, analysis, and visualization are shown (Available: ✓; Limited availability: L; Not available: ✗).

## Results

**Mesmerize allows for rich data annotation**. The first step of any calcium imaging analysis workflow requires a system for users to explore their imaging data and perform ROI extraction. We demonstrate that Mesmerize works with both 2D and 3D datasets from a broad set of model organisms, such as mice, zebrafish, and *Ciona intestinalis* (Fig. 1a). These datasets can be visualized using the Mesmerize Viewer, which provides GUI front-ends (based on pyqtgraph) and API interfaces for various signal extraction modules (Fig. 1b). Importantly, the Viewer also facilitates extensive in-place annotation of experimental information (Fig. 1c–e), such as but not limited to

1. temporal mapping, such as stimulus or behavioral periods (Fig. 1d);
2. cell identities, morphology, or any other tags that map to individual cells/ROIs (Fig. 1e).

These annotations may be performed through the GUI, or automated through the simple scripting interface. Mesmerize's unique support for customizable annotations makes it broadly applicable for a diverse range of researchers and distinguishes it from other calcium imaging and image analysis tools (Table 1). The highly versatile annotation functions within Mesmerize enable scientists to efficiently curate and analyze complex datasets that are emerging from the use of multiplexed imaging combining several cell-specific promoters that express Genetically Encoded Calcium Indicators (GECIs). For example, researchers can perform a cohort of experiments that utilize tens of GCaMP promoters, multiple combinations of optogenetic and/or chemogenetic lines, multiple UAS-GAL4 systems, multiple drugs etc. in one efficient, organized and reproducible system. To illustrate this capacity of Mesmerize, we leverage a powerful emerging model organism, the protochordate *C. intestinalis*. The *Ciona* dataset analyzed here includes annotations for seven different GCaMP6s promoters, eight anatomical regions, and 21 cell types (Supplementary Tables 1 and 2).

**ROI extraction**. Graphical front-ends help users explore imaging data, perform preprocessing, and signal extraction. They help facilitate efficient workflows for advanced users and are necessary for users without extensive programming experience. From a user's perspective these front-ends, which we call Viewer Modules, interact with the Mesmerize Viewer in a manner similar to the various components within ImageJ and its plugins. This familiarity in the user-end design will allow Mesmerize to be easily adopted by more biologists, and broaden the reach of cutting-edge packages (such as the CaImAn library[1]), allowing users to perform more accurate and in-depth analysis.

By default, Viewer Modules are provided for NoRMCorr[22], CNMF(E)[23,24], NuSeT[25], as well as importers for Suite2p[2] outputs and ImageJ[5] ROIs (Fig. 1b). These front-ends encompass a very broad variety of user-options for motion correction and signal extraction from both 2D and 3D calcium imaging datasets. Many Viewer Modules are used in conjunction with the Mesmerize Batch Manager which streamlines the exploration of parameter space and data organization for these computationally intensive tasks.

ROI extraction and image processing are not limited to the default options that we provide; these Viewer Modules can be expanded, customized and created by users with modest programming experience. We provide an API and scripting interfaces, which allows ROIs to be extracted from any other custom technique which the user may desire. This flexibility allows scientists to conveniently integrate and combine their

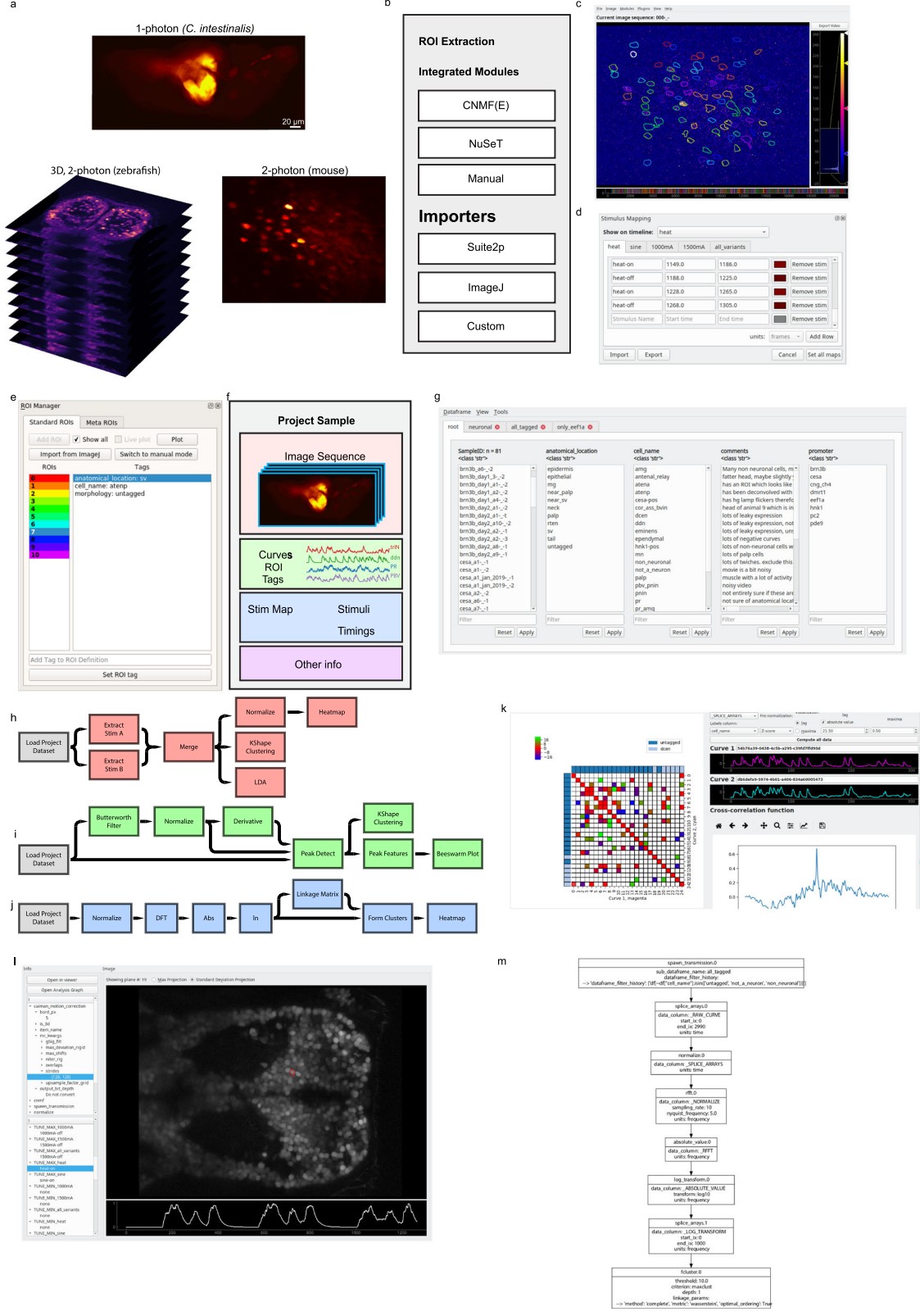

favorite preprocessing or ROI extraction technique into their analysis workflow. For example, we created a simple API[26] to a deep-learning approach for cellular segmentation using the NuSeT[25] network, which is useful for the segmentation of recordings using nuclear-localized GCaMP. The NuSeT method can be used through a GUI that can be expanded to include additional deep-learning segmentation approaches from this

rapidly evolving field in the future. Furthermore, the binary masks produced by the NuSeT Viewer Module can be used for seeding CNMF(E)[23,24], thereby allowing these two cutting-edge tools to be combined in manner that would be non-trivial for users without extensive programming experience. In summary, these features demonstrate how Mesmerize can be a powerful platform for complex integration and interoperability between

**Fig. 1 Mesmerize platform overview. a** Raw imaging data that can originate from a variety of sources; examples shown from 1-photon calcium imaging of *Ciona intestinalis*, 2-photon imaging of the mouse visual cortex neurons and volumetric 2-photon imaging of zebrafish. **b** Mesmerize's highly modular design allows ROI extraction to be performed through a variety of methods such as CalmAn CNMF(E), NuSeT deep learning, or manually. ROIs can also be imported from Suite2p, ImageJ, or a custom module can be written using the API to import ROIs from other sources. **c** The Mesmerize Viewer lets users explore their imaging data and integrates with various viewer modules such as: **d** Stimulus Mapping module which allows users to map temporal information, such as stimulus or behavioral periods; **e** ROI Manager that can work with ROIs originating from a variety of sources, as shown in (**b**), and allows users to tag an unlimited variety of categorical information such as anatomical location, cell type, morphology, etc. for each ROI. **f** All data pertaining to an imaging session, i.e. the image sequence, calcium curves, ROIs, tags (annotations), stimulus mappings, and all other categorical information are packaged into a Project Sample and saved to the Project Dataset. **g** The samples within a Project Dataset can be interactively managed using the Project Browser. **h–j** Project Datasets, or sub-datasets, can be loaded into a flowchart to interactively perform downstream analysis. Simplified examples of how flowcharts can be used to **h** explore stimulus or behavioral responses, **i** analyze peak features (width, amplitude, slope etc.) or perform *k*-shape clustering and **j** perform hierarchical clustering. **k** Downstream analysis in flowcharts are integrated with various forms of highly interactive plots such as cross-correlation analysis. Many interactive plots are associated with a **l** Datapoint Tracer where users can click on individual datapoints to view the spatial location of the ROI that it originates from, along with all other data associated with that datapoint. **m** The Datapoint Tracer shown in (**l**) also lets users view the analysis history log for every datapoint in the form of an Analysis Graph.

multiple state-of-the-art analysis tools for both end-users and developers.

**Project organization**. Current software platforms for bio-image dataset organization are not suited for handling calcium imaging data (Table 1). Mesmerize packages all data associated with an imaging sample, i.e. extracted signals, annotations etc., into a Project Sample (Fig. 1f). A collection of Project Samples constitute a Project Dataset, which can be explored and filtered in a user-friendly manner to create experimental groups using the Project Browser (Fig. 1g). Project Samples can be modified throughout the course of a project. Therefore, in addition to efficient data annotation, users can append, change, or supplement existing annotations that can then be propagated through downstream analysis and visualizations. Dynamically adaptable data management is extremely useful since biological questions and experiments are often in constant flux as new data are processed and analyzed.

**Downstream analysis**. A Project Dataset, or sub-dataset, can be loaded into a flowchart where users can build analysis pipelines by connecting analysis nodes (Fig. 1h–j). We provide nodes to perform many common signal processing routines, data handling/organization, dimensionality reduction, and clustering analysis. Mesmerize's default collection of nodes allows users to perform many common analysis procedures such as comparison of stimulus/behavioral periods (Fig. 1h), peak detection (Fig. 1i), and clustering analysis (Fig. 1j). All analyses performed in the flowchart are logged with a description of the nodes and their parameters, thereby facilitating future reproducibility of the analyses. For more customized analysis, we provide documentation and an API for efficiently writing new analysis nodes or using the analysis data structures in external notebooks or scripts (http://docs.mesmerizelab.org/en/master/developer_guide/nodes.html). The flowchart builds upon a pyqtgraph[27] widget. The stock assortment of nodes implement various signal processing, dimensionality reduction, and clustering analysis using scipy[28], sklearn[29], and tslearn[30] libraries. We use common and mature libraries to simplify customization by more advanced users or developers.

**Visualization**. The ultimate result of almost any analysis procedure and scientific study is the creation of visualizations that convey an experiment's results. The vast majority of visualizations in most research are static. This makes it difficult or impossible to instantly link datapoints from a plot with the original imaging data and analysis procedures[8,9,16], which greatly hampers

reproducibility[16]. Recent developments help address these issues; tools such as Jupyter[31] notebooks delivered via MyBinder[32] allow the data and analysis procedures to be shared. However, these methods are not readily accessible to non-programmers and do not aid in the creation of FAIR and functionally linked datasets. Mesmerize allows users to create interactive visualizations through a GUI and share them in their interactive state (Fig. 1k). Many interactive plots are attached to a *Datapoint Tracer* (Fig. 1l), which highlights the spatial localization of the selected datapoint and displays all its associated annotations and the analysis history log which can be visualized using an analysis graph (Fig. 1m), a graphical visualization that intuitively communicates the analysis steps. A rich variety of built-in plots are provided, such as heatmaps, spacemaps, scatterplots, beeswarm, and more. As with other components of the Mesmerize platform, we provide developer instructions for the creation of new plots that can integrate with the Datapoint Tracer (http://docs.mesmerizelab.org/en/master/developer_guide/plots.html). Thus far, no other calcium imaging analysis suite offers such a rich variety of interactive visualizations for downstream analysis (Table 1). Lastly, we are currently creating a set of standardized web-based visualizations that mirror the current options available for matplotlib[33] and pyqtgraph[27] based plots in Mesmerize. This will further improve the shareability of data since a user will be able to interactively explore visualizations from a Mesmerize dataset without installing anything on their end.

**Shareable datasets**. In summary, Mesmerize is the first platform to address common difficulties with reproducibility, data reusability, and organization in calcium imaging data analysis by comprehensively encapsulating image analysis, data annotation, downstream analysis, and interactive visualizations. Mesmerize allows analysis procedures and annotations to be transparent at the level of individual datapoints in a plot. This is achieved by tagging Universally Unique Identifiers (UUID) to the data at various layers of analysis, a key principle for the creation of a FAIR dataset. Mesmerize's unique capacity for the robust maintenance of rich and complex annotations encourages users to exhaustively describe their datasets. A Mesmerize project is entirely self-contained within a single directory tree, making it easy to share entire datasets, analysis workflows, and interactive visualizations with the scientific community. Another scientist can open a Mesmerize project and immediately explore visualizations, analysis procedures, and view the raw data associated with the datapoints on a published figure. This ease of opening a Mesmerize project and exploring datasets in conjunction with interactive visualizations will help scientists in making their data easily accessible and reusable.

Lastly, in order to reach a broad range of users, Mesmerize is cross-platform and works on Linux, Mac OSX, and Windows. Mesmerize is free, open source, uses the GNU General Public License v3.0 and is hosted on GitHub. To facilitate fast and easy installation on all major platforms, we provide an importable Virtual Machine with Mesmerize pre-installed so that users can get up and running within minutes. Mesmerize is also on PyPI, which allows it to be installed via pip—the prevailing package manager for Python. We have a dedicated YouTube channel with more than 150 min of video tutorials, we host an active GitHub community to provide troubleshooting help, software maintenance, and a gitter room for open discussions. Mesmerize is regularly updated and there have been five releases in the past year (excluding bug-fix releases). This paper describes Mesmerize v0.7.1. See the "Code availability" section for details.

**Calcium imaging in the mouse visual cortex in response to visual sinusoidal grating stimuli**. Before we illustrate the more complex and novel analysis that can be performed with Mesmerize, we demonstrate its use for basic neurobiological analysis using a well-known phenomenon and a simple dataset. We used a mouse visual cortex dataset (dataset name: CRCNS pvc-7) contributed by the Allen Brain Institute, which consists of in vivo 2-photon imaging data from layer 4 cells in the mouse visual cortex[34] (Fig. 2a). The recording was performed while the mouse was presented with visual stimuli consisting of sinusoidal bands at various orientations, spatial frequencies, and temporal frequencies. The stimulus mapping module in Mesmerize allows users to map temporal annotations, such as the characteristics of the visual stimuli in this experiment (Fig. 2b). However, it can be used to map any temporal variable, such as behaviors and other forms of stimuli, with any number of characteristics. These temporal mappings can be entered manually through the GUI, or the scripting interface can be used to import a temporal mapping from a spreadsheet file. As we will show, these temporal mappings can be incorporated into the downstream analysis—an essential feature for streamlined analysis in systems neuroscience. The CaImAn NoRMCorre[22] module and CNMF[23] were used for motion correction and signal extraction respectively (Fig. 2c). A flowchart, illustrated in Fig. 2d, can then be used to determine how cells are tuned to various characteristics of the visual stimuli. An interactive heatmap can be used to visualize the result (Fig. 2e). The heatmap can be labeled and sorted according to any categorical variable in the dataset, such as the orientation, spatial frequency, and temporal frequency that each cell is tuned to. As mentioned previously, clicking a datapoint in the heatmap will update the Datapoint Tracer, which then (1) highlights the spatial localization of the ROI that the datapoint originates from, (2) displays all other data associated to the datapoint (Fig. 2e, bottom center), and (3) lists the analysis log (Fig. 2e, top center) which can be exported as an analysis graph (Supplementary Fig. 1). Another visualization that is appropriate for these data are Spacemaps. These allow users to spatially visualize categorical analysis results or annotations within the imaging field. For example, we show orientation tuning (Fig. 2f), spatial frequency tuning (Fig. 2g), and temporal frequency tuning (Fig. 2h) of the cells in the CRCNS pvc-7 dataset. The analysis of this basic dataset illustrates how Mesmerize can encapsulate entire analysis workflows.

**Analysis of a volumetric zebrafish calcium imaging dataset coupled to somatosensory stimulation**. Mesmerize is also capable of handling 3D volumetric imaging datasets with the same annotation and analysis capabilities that are provided for 2D datasets. In order to demonstrate some of these features we analyzed an in vivo 2-photon imaging dataset where zebrafish larvae expressing a nuclear-localized GCaMP are presented with various forms of heat stimuli[35] (Fig. 3a). Users are provided with multiple options for ROI extraction from 3D data. Mesmerize can interface with the Caiman 3D CNMF[23] implementation, or each plane can be processed individually using Caiman 2D CNMF. Furthermore, Mesmerize can utilize the NuSeT[25] network to provide a deep-learning-based segmentation tool for ROI extraction. These NuSeT-segmented ROIs that can then be used to initialize CNMF. This example demonstrates how Mesmerize's modular platform greatly simplifies the process of combining multiple cutting-edge tools, allowing them to be more easily adopted by a broader range of users. For this 3D dataset, CNMF with greedy initialization performed poorly (Fig. 3b), which is likely due to lower signal-to-noise ratios that are more common with 2-photon volumetric imaging[36]. However, the performance of CNMF is greatly improved when it is initialized with binary masked produced by NuSeT (Fig. 3b). After ROI extraction, the stimulus information was temporally mapped and a few imaging samples were used to create a Mesmerize project and perform downstream analysis. Interactive stimulus tuning plots can be obtained for every cell (Fig. 3c, d), and these can be used to sort cells according to the stimulus they are tuned for (Fig. 3e) and visualized using a spacemap (Fig. 3f). Lastly, we used Mesmerize to train a linear discriminant analysis (LDA) model and classified three distinct brain states that are observed during heat-on, heat-on-delayed, and pre-stimulus (none) periods (Fig. 3g). Put together, these demonstrate Mesmerize's capabilities in handling 3D calcium imaging data and identifying distinct brain states using standard machine learning approaches, such as LDA. This example demonstrates how Mesmerize's suite of analysis tools and annotation capabilities makes it a game-changer for cutting-edge systems neuroscience researchers in the present and into the future as volumetric imaging becomes more widespread.

**Functional fingerprinting of neuronal and non-neuronal cell types in *C. intestinalis***. Having demonstrated how Mesmerize can be used to tackle several popular experimental paradigms in neuroscience, where neuronal dynamics are analyzed in the context of stimuli or behavior, we next addressed more contemporary/non-standard forms of analysis, with the aim of making novel biological findings. We thus turned our attention to spontaneous calcium activity datasets from both neuronal and non-neuronal cells in the absence of well-defined stimuli, in cells where typical neuronal spike trains have not been observed previously by leveraging the emerging model organism for systems neuroscience, the protochordate *C. intestinalis*. Neurobiological studies in *C. intestinalis* have just gained momentum, with a handful of ethological studies[37–39] and a few studies of calcium dynamics[40]. However, no pan-neuronal calcium imaging analysis has been performed and such a study would be a great resource for the *Ciona* and greater chordate community.

We chose *C. intestinalis* as a model system to address the unique and fundamental question of spontaneous neuronal activity in neuronal and non-neuronal cells for multiple reasons. First, the recent completion of the larval connectome[41–43] in conjunction with the generation of comprehensive single-cell transcriptomes[44,45] establishes the nervous system of *C. intestinalis* as likely the most thoroughly mapped chordate nervous system to date. Second, despite the established connectome, there has not been a comprehensive functional study to investigate neuronal activity across its diverse neuronal populations. Third, its small nervous system, flat head, and the ability to label genetically defined populations of cells using various promoters that drive GCaMP6s expression allow us to approximate the

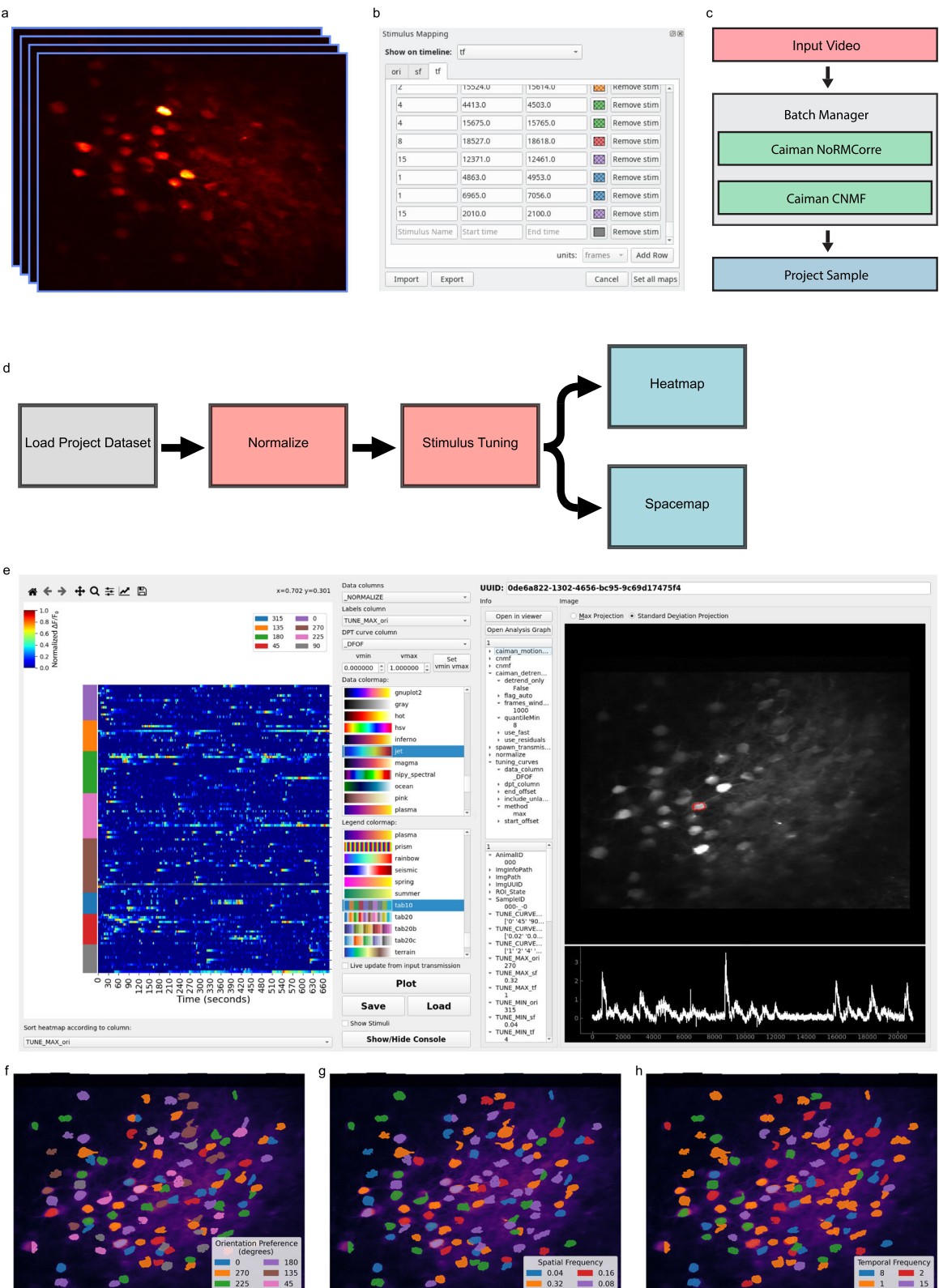

identity of neuronal cells in reference to the connectome[41,42]. Finally, to showcase comprehensive comparative calcium dynamics analysis within the same organism for applications beyond neuroscience, we additionally performed calcium imaging in two non-neuronal cell types in *C. intestinalis*, the epidermis and a population of migratory mesenchymal cells termed trunk lateral cells[46] (TLCs). The analysis methods developed in this work can be employed by cell and developmental biologists to study calcium-dependent mechanisms that underlie a broad range of cell biological and morphogenetic processes.

Since our goal here was to quantitatively define calcium activities in cells and domains where typical neuronal spike trains have not been observed previously, we implemented techniques which have not been used prior to our study to analyze calcium

**Fig. 2 Stimulus tuning of cells from the CRCNS PVC-7 dataset. a** Video of cells within the visual cortex of a mouse being presented with visual stimuli consisting of sinusoidal gratings. These stimuli can be mapped onto the imaging data using the **b** Stimulus Mapping module of the Mesmerize Viewer. **c** The video was processed using the Mesmerize Batch Manager, which allows users to conveniently manage computationally intensive tasks such as CalmAn NoRMCorre motion correction and CNMF(E). The CNMF results are imported in the Mesmerize Viewer and are packaged into a Project Sample with the imaging data and stimulus maps. **d** Flowchart that illustrates basic stimulus tuning analysis that can be performed in Mesmerize flowcharts. **e** Heatmap widget showing the results of the stimulus tuning analysis flowchart in (**d**). The heatmap shows min-max normalized calcium traces. The y axis color labels show the orientation tuning of the cells. These plots are interactive, allowing the user to plot various forms of numerical data, such as raw traces, normalized traces, $\Delta F/F_O$, z-scored traces etc., the relationships between numerical data and various form of categorical data such as stimulus tuning, ROI tags, etc. The spatial location of the ROI and calcium trace, along with any other tagged data, can be seen on the right-hand-side panels of the widget (Datapoint Tracer). The stimulus tuning of individual cells can also be visualized using Spacemaps to visualize the (**f**) orientation tuning of cells, **g** spatial frequency tuning, and **h** temporal frequency tuning. Spacemaps can be used to visualize ROIs with respect to any categorical variables.

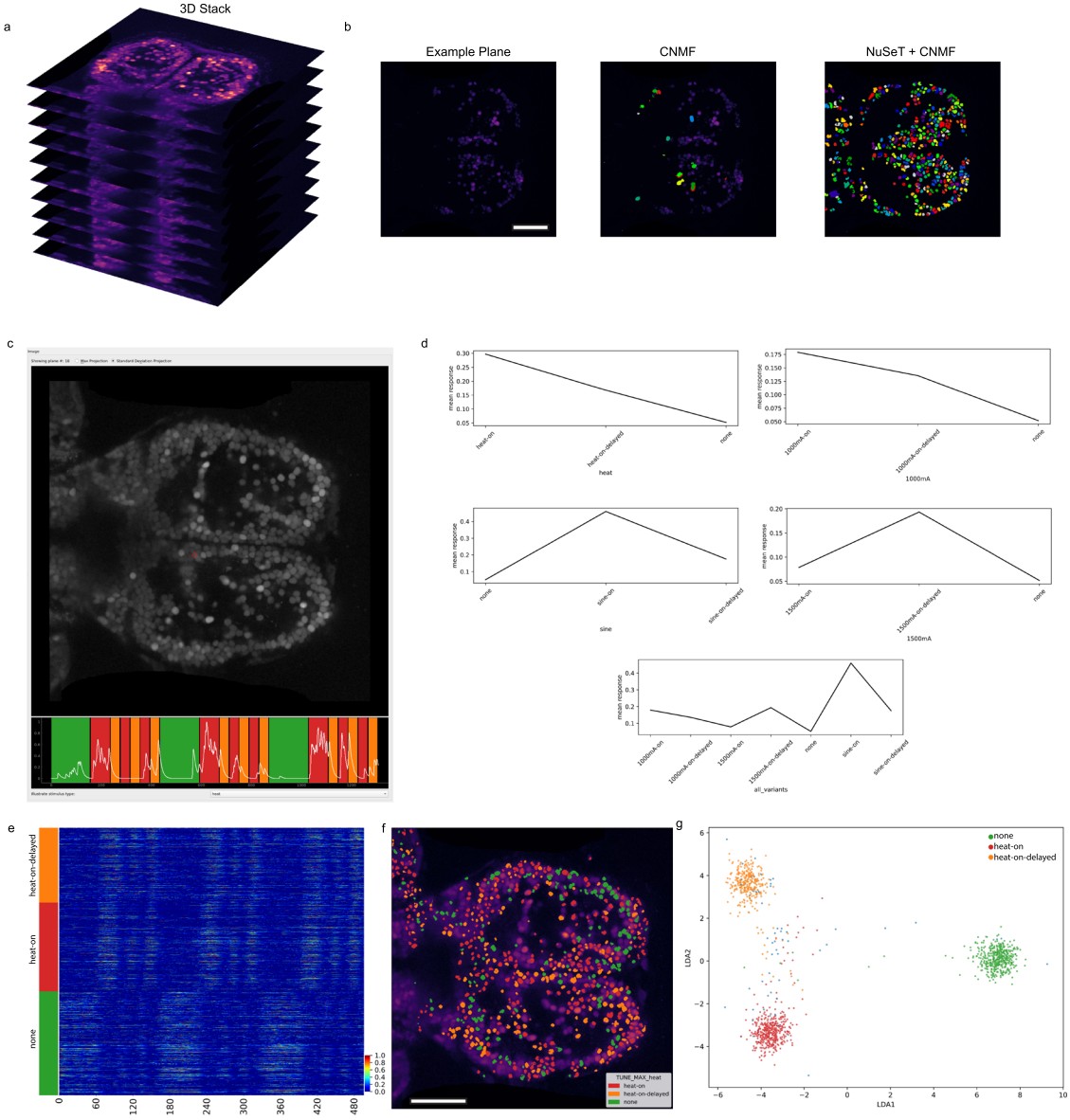

**Fig. 3 Mesmerize handles 3D calcium imaging data. a** Mesmerize can work with volumetric calcium imaging data. **b** Frame from one plane of the volumetric dataset. CNMF with greedy initialization is unable to detect many cells in this relatively noisy dataset; however, CNMF seeded with NuSeT segmentation picks up many more cells. **c** Mesmerize datapoint tracer showing a cell highlighted in red, and the corresponding calcium trace. The tuning curves of this cell are shown in (**d**), which shows that this cell is tuned to heat-on stimulus. **e** Min−max normalized calcium traces sorted by their stimulus tuning profiles, heat-on, heat-on-delayed, and none. Color bar indicates normalized fluorescence intensity. **f** Spacemap showing the stimulus tuning characteristic of each cell. **g** LDA projection showing distinct brain states for heat-on, heat-on-delayed and none between each stimulus trial. Scale bars: 100 μm. heat-on: cells that respond to the heat stimulus; heat-on-delayed: cells that show a delayed response to the heat stimulus; none: cells that are more active between the stimulus trials and less active during heat-on and heat-on-delayed stimulus periods.

dynamics. These methods can also be applied to understand calcium dynamics in other systems. Frequency-domain analysis has previously been used to compare calcium dynamics between experimental groups[47,48] and during cortical development[49]; however, it has not been used for global clustering analysis to deduce more complex relationships between cell types or experimental conditions. To fill this gap, we introduce the application of Earth Mover's Distances[50,51] (EMD) between frequency-domain representations of calcium traces data as a distance metric for hierarchical clustering. The EMD is commonly used for pattern recognition and image retrieval systems through histogram comparison[51]. Intuitively, the EMD can be thought of the amount of work that must be done to transform one distribution into another. Therefore, in contrast to the Euclidean distance, the EMD accounts for the order of elements along two feature vectors that are being compared. This makes it a useful metric for performing clustering analysis using discrete Fourier transforms (DFTs) of calcium traces since similar weights in neighboring, but not identical, frequency domains are measured as a small EMD whereas the same weights in far-apart frequency domains result in a large EMD between the feature vectors. To illustrate this, consider the traces from two cells that appear to have similar dynamics (Fig. 4a), and their corresponding Fourier transforms (Fig. 4b). If the order of elements along the DFT, shown as feature vectors $u$ and $v$ (Fig. 4b), are randomly shuffled, the EMD between the shuffled vectors is different whereas the Euclidean distance is identical (Fig. 4c).

Next, we show how we used the EMD to cluster calcium dynamics of neuronal and non-neuronal cells from *C. intestinalis*. To conceptually demonstrate the application of EMD, consider eleven example traces (Fig. 4d). It is important to note that these traces were not acquired over the same time period and we were not interested in finding neurons/cells that fire together (i.e. neural assemblies). Instead, we were interested in quantitatively categorizing neurons based on their overall dynamics. The EMD-based distance matrix shows better grouping than the distance matrices calculated using Euclidean distances (Fig. 4e, f). To quantitatively demonstrate that the EMD performs better than Euclidean distances, we performed hierarchical clustering and calculated the agglomerative coefficient (denoted by $\alpha$)—a score between 0 and 1 where values approaching 1 indicate better clustering structure. With the ten example traces, the hierarchical clustering obtained by using the EMD metric results in an agglomerative coefficient $\alpha \approx 0.841$ (Fig. 4g), whereas the clustering obtained from Euclidean distances results in a coefficient $\alpha \approx 0.574$ (Fig. 4h). When applied to a larger dataset the clustering structure found through EMD is even stronger with an agglomerative coefficient $\alpha \approx 0.983$ (Fig. 4i), compared to $\alpha \approx 0.663$ for Euclidean distances (Fig. 4j). Agglomerative coefficients tend to increase with the size of a dataset; therefore, smaller datasets (Fig. 4e, f) are more useful for evaluating performance between different metrics. Euclidean distances in the time domain can be useful for grouping cells that fire together; however, this is irrelevant since the traces were not acquired over the same time period.

To compare our methods with techniques that have previously been used in clustering analysis of spontaneous neuronal activity, such as comparisons between various stages of the circadian cycle[52], we benchmarked Silhouette and Davies−Bouldin scores using both hierarchical and $k$-means clustering. EMD-based hierarchical clustering far outperforms standard hierarchical clustering using Euclidean distances, and $k$-means using both the time and frequency domain (Fig. 4k, l). Since the data are not temporally aligned, $k$-means clustering would be unsuitable for our task and mostly results in aligned traces as expected (Supplementary Fig. 2). From these dendrograms and agglomerative coefficients, we demonstrate that the EMD metric between frequency-domain representations of calcium traces results in

better separation of disparate dynamics and an aggregation of similar dynamics. Since this method is suitable for data that are not temporally aligned, it opens the potential for novel analysis of spontaneous activity during circadian cycles[52], development[49], and during pathological states using psychiatric disease-relevant models and paradigms[48,53].

To illustrate how the EMD is a simple and effective method for characterization of calcium dynamics across a diverse range of cell types, we performed hierarchical clustering on traces obtained by imaging various neuronal and non-neuronal populations of cells in the *C. intestinalis* head. Clustering of both neuronal and non-neuronal cells resulted in a dendrogram which was cut to form four clusters, separating these cells into four distinct populations based on their activity profile (Fig. 5a). Example traces from each of the four clusters show that cluster 1 consists of cells with very low levels of activity (Fig. 5b). Cells within cluster 2 show slightly more activity, and cluster 3 is enriched with cells showing moderately more activity and shorter peaks. Cluster 4 is highly enriched with cells that show very high levels of activity. The cluster centroids help to further describe the characteristics of the four clusters. Cluster 1 shows very high spectral energy in the lowest frequency domains, and relatively no spectral energy in higher frequency domains (Fig. 5c). The amount of spectral energy in the lowest frequency domains decreases progressively from cluster 1 to cluster 4, whereas the opposite is true for spectral energy in higher frequency domains. Cluster 4 shows the most spectral energy in higher frequency domains. Biologically, each of these four clusters are enriched with distinct populations of cells (Fig. 5d). Cluster 1 is almost exclusively composed of CESA and HNK-1 cells exhibiting wide and large peaks, with high spectral energy in lower frequency domains. In contrast, neuronal cells are predominantly found in clusters 3 and 4, with a few peripheral sensory neurons also found in cluster 2. Peripheral sensory neurons, such as Palp, aATEN, pATEN and RTEN, are highly enriched in clusters 2 and 3. Cluster 4, with cells showing very high activity, mostly consists of various types of photoreceptor cells and interneurons.

This analysis demonstrates that the combination of DFT with EMD allows us to identify different activity states in non-neuronal cell types and to classify different neuronal cell types in different groups based on their activity dynamics. We show that this clustering separates genetically defined populations of peripheral and sensory neurons, from populations located within the brain vesicle which form the central nervous system. Most interestingly, four cell types involved in peripheral sensory networks namely the Palp Sensory Neurons (PSNs), the rostal trunk epidermal neurons (RTEN), and the apical trunk epidermal neurons (aATEN & pATEN) exhibit similar modes of activity and are enriched in clusters 2 and 3. Previous anatomical studies[43,54,55] postulated that PSNs provide feedforward excitation to the RTENs, while all four cell types appear to exhibit a glutamatergic molecular signature[54,56]. The similarity in their activity signatures that we observe in our imaging analysis provides functional support for this hypothesis. Cells that are mostly primary interneurons within the brain vesicle all exhibit high levels of activity and cluster together (Fig. 5d). These cell types include interneurons that are postsynaptic to the RTENs such as the peripheral interneurons (PNIN), interneurons closely associated with photoreceptors such as the photoreceptor tract interneuron (trIN) and the photoreceptor relay neurons (prRN), antenna relay neurons (antRN) which receive input from the gravity sensing cells and finally the Eminens (Em) peripheral relay neurons which are thought to be one of the main centers of integration in the larval nervous system based on the number of synaptic partners they have[41]. In agreement with the emerging view from the larval connectome, the high activity that these

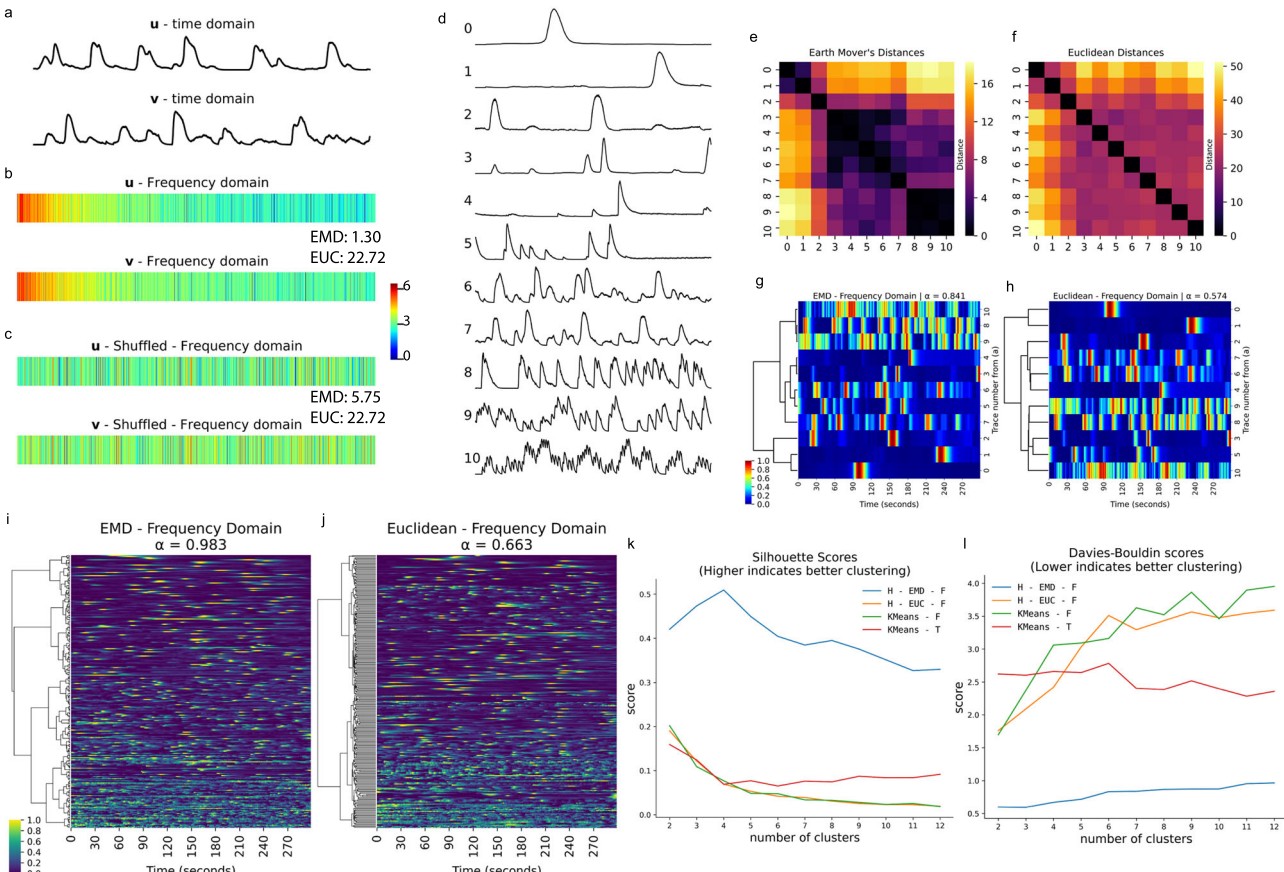

**Fig. 4 The Earth Mover's Distance is a robust metric for broadly characterizing calcium activity. a** Two example calcium traces, *u* and *v*, in the time domain. **b** Discrete Fourier transforms (DFTs) of *u* and *v* are used as feature vectors. The Earth Mover's Distance (EMD) between *u* and *v* is 1.30, the Euclidean (EUC) distance between *u* and *v* is 22.72. **c** A random shuffle is applied to feature vectors *u* and *v*. The Earth Mover's Distance (EMD = 5.75) changes after the random shuffle; however, the Euclidean distance (EUC = 22.72) is identical. This demonstrates how the order of elements along a feature vector is captured by the EMD, which is necessary for effectively comparing DFTs. Color bar for (**b**) and (**c**) indicates square root of energy. **d** Eleven example calcium traces from *C. intestinalis*. **e** Distance matrix showing EMDs between DFTs of the 11 calcium traces from (**d**). **f** Distance matrix showing Euclidean Distances between DFTs of the 11 calcium traces from (**d**). **g** Dendrogram constructed from (**e**), with a high agglomerative coefficient ($\alpha \approx 0.841$, best = 1, worst = 0) indicating good hierarchical clustering. **h** Dendrogram constructed from (**f**), with a low agglomerative coefficient ($\alpha \approx 0.574$), indicating poor hierarchical clustering. Color bar for (**g**) indicates normalized fluorescence intensity for (**g**) and (**h**). **i, j** Dendrograms showing hierarchical relationships between over 200 calcium traces. Color bar indicates min-max normalized fluorescence. **i** Dendrogram calculated using EMD, showing a very high agglomerative coefficient ($\alpha \approx 0.983$) that indicates good clustering performance. Cells near the top of the tree show slow and sparse calcium dynamics, cells closer to the bottom of the tree show much more active and complex calcium dynamics. **j** Dendrogram calculated using Euclidean distances, showing a moderate agglomerative coefficient ($\alpha \approx 0.663$). **k** Silhouette scores comparing clustering performance of various methods, higher scores indicate better, performance. Hierarchical clustering using the EMD between DFTs outperforms other methods. **l** Davies−Bouldin scores comparing clustering performance of various methods; lower scores indicate better clustering performance. This score also demonstrates that hierarchical clustering using the EMD between DFTs outperforms other methods. Abbreviations from (**l**) are defined in Supplementary Table 2.

different types of interneurons exhibit likely reflects the more complex inputs that they receive due to their intermediate positions in different sensory networks.

The distinct clustering of cell types shown here is likely indicative of cellular function and molecular composition. For example, the slower calcium dynamics observed in cluster 1 likely reflect the contribution of calcium signaling in homeostatic cellular processes[57] such as epidermal barrier formation and maintenance, and processes mediating motility and cell-shape changes in mesenchymal cells. Neuronal cells are inherently noisy compared to other excitable cell types[58], such as epithelial cells, even in the absence of any discernable stimuli. However noise, or spontaneous activity, is often important for many neurobiological processes such as development[49], encoding[59] and stochastic resonance[60–63]—a signal-boosting strategy employed by sensory circuits and other neurophysiological systems where noise from

neurons exhibiting spontaneous activity is injected to increase the sensitivity of sensory circuits. Spontaneous activity in developing circuits have been studied semi-quantitatively, including frequency analysis[49]. These fields could greatly benefit from a method to quantitatively compare and cluster large numbers of diverse cell types to create cell-type signatures at various stages of development, which could complement the ever-growing transcriptomic data that are more commonly used to generate cell-type signatures[64]. Put together, this work reveals how spontaneous activity is sufficient to broadly derive cell-specific functional fingerprints in *C. intestinalis* larvae. This simple but broadly applicable technique can be used in other model systems to define discrete functional domains for specific populations or sub-types of neurons and provides a novel way to quantitatively characterize the overall dynamics of calcium, or other molecules and ions.

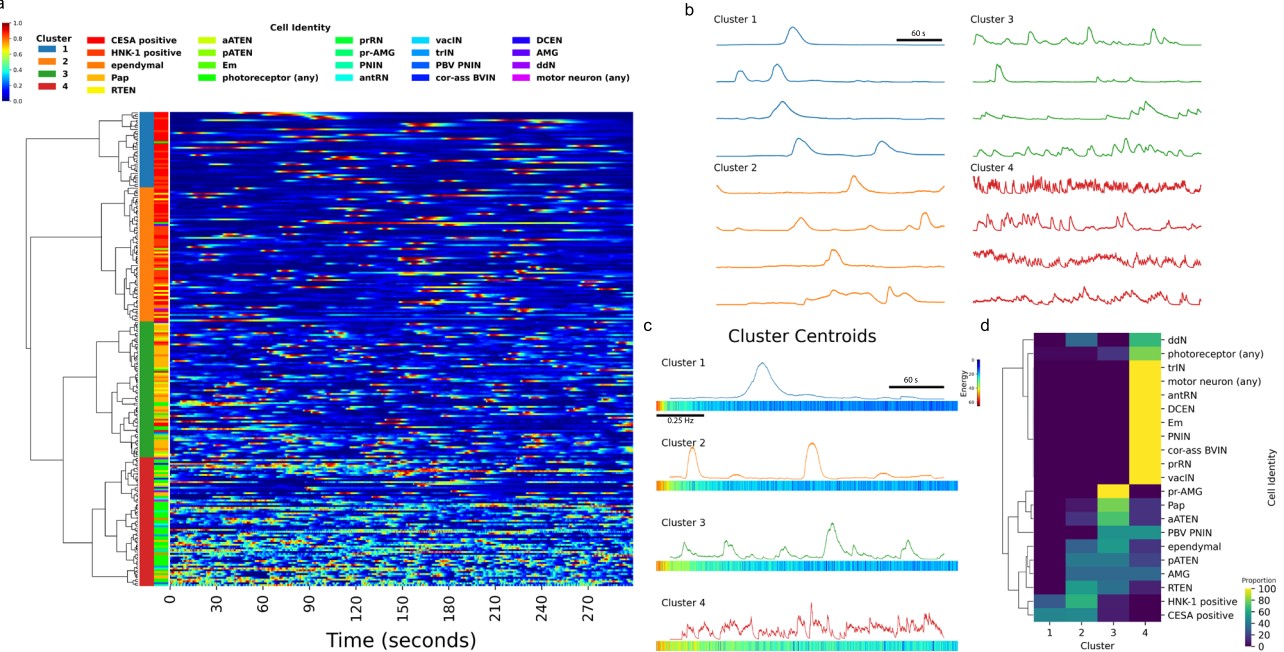

**Fig. 5 Spontaneous calcium dynamics in *C. intestinalis* reveals cell-type signatures. a** Hierarchical clustering of calcium dynamics observed in neuronal and non-neuronal cells within the head of *C. intestinalis*. Dendrogram shows hierarchical relationships. Top left color bar indicates the min−max normalized fluorescence intensity scale used for the heatmap. Left color bar legend between the dendrogram and heatmap indicates cluster membership. Right color bar legend indicates cell identity. Heatmap shows normalized traces. **b** Example traces from each cluster. **c** Cluster centroids in both the time domain (top) and frequency domain (bottom). **d** Proportion of cells that appear in each of the four clusters. For each cell type, proportions sum up to 100% across all four clusters. Cell-type abbreviations are defined in Supplementary Table 3.

**Motif extraction from shape-based analysis of calcium imaging data.** To extract additional valuable information from our calcium imaging datasets, here we demonstrate another downstream analysis method, k-Shape clustering[30,65], on our *C. intestinalis* dataset using Mesmerize. Many experiments in neuroscience and cell biology require a quantitative method to define discrete archetypical shapes from calcium traces, as well as traces that may represent changes in the levels of other molecules such as those obtained from neurotransmitter or voltage indicators, etc. Thus, the methods described here will be broadly applicable to trace-containing datasets and not limited to calcium datasets. In the early days shape archetypes were defined subjectively[66–69], and currently the most common method is to describe peak-features such as amplitude, width, slope, etc.[70]. However, certain biological systems such as the developing nervous system or adult nervous system in the context of pathological conditions (e.g. seizures) display complex and irregular types of calcium activity, which makes the use of such metrics less suitable. Here we apply k-Shape clustering, a contemporary time-series analysis technique to tackle this problem. This method allows us to comprehensively compare peaks directly so that we can reduce calcium traces to sequences of discrete motifs. k-Shape clustering uses a normalized cross-correlation function to derive a shape-based distance metric that can be used to extract a finite set of discrete archetypical peaks from calcium traces (Fig. 6a). These clusters can be visualized using PCA of peak features to illustrate how the k-Shape clustering maps to more traditional peak-features based measures (Fig. 6b–c). k-Shape derived archetypes can then be used to reduce calcium traces to sequences of discrete letters, and statistical models, such as Markov chains (Fig. 6d–g), can be applied to describe calcium dynamics between different types of cells or experimental groups. For example, the Markov chains created using k-Shape-sequences derived from HNK-1 traces (Fig. 6d, e) are very simple, characteristic of the simple calcium dynamics that these cell exhibit. On the other hand, Markov chains that represent photoreceptor cells

(Fig. 6f, g) are much more complex. In summary, we show that k-Shape clustering could provide a contemporary approach to answering questions in various systems, such as examining stimulus-response profiles, behavioral periods, etc. This approach can likely be further tailored to extract motifs from imaging calcium, neurotransmitters, voltage or other Genetically Encoded Indicators (GEIs) using different organisms, to investigate conserved and species-specific mechanisms.

## Discussion
We demonstrate here that Mesmerize is a platform that can be used to perform novel, complex, and reproducible calcium imaging data from a diverse range of cell types and organisms.

Mesmerize addresses a contemporary need in the field of functional imaging namely, the requirement for a platform with cutting-edge analytical tools capable of tackling 2D and 3D datasets that is accessible to biologists with a broad range of competence in terms of computational skills and biological interests. We show that Mesmerize can analyze a wide range of datasets from multiple organisms with morphologically diverse brains and cell types, which were acquired using different imaging techniques (e.g., 2-photon imaging, epifluorescence) in the absence or presence of spatiotemporally defined external stimuli.

While the creation of a user-friendly platform was of paramount importance, this should not come at the expense of novelty, expandability, traceability, and broad applicability. Mesmerize provides new analysis techniques such as EMD-based hierarchical clustering and k-Shape clustering in combination with Markov chains, equipping users with new tools to extract functional fingerprints and to delineate the basic building blocks and organization of calcium activity from diverse cell types. Our platform can be readily integrated with popular imaging processing tools such as Suite2p and can utilize newly published

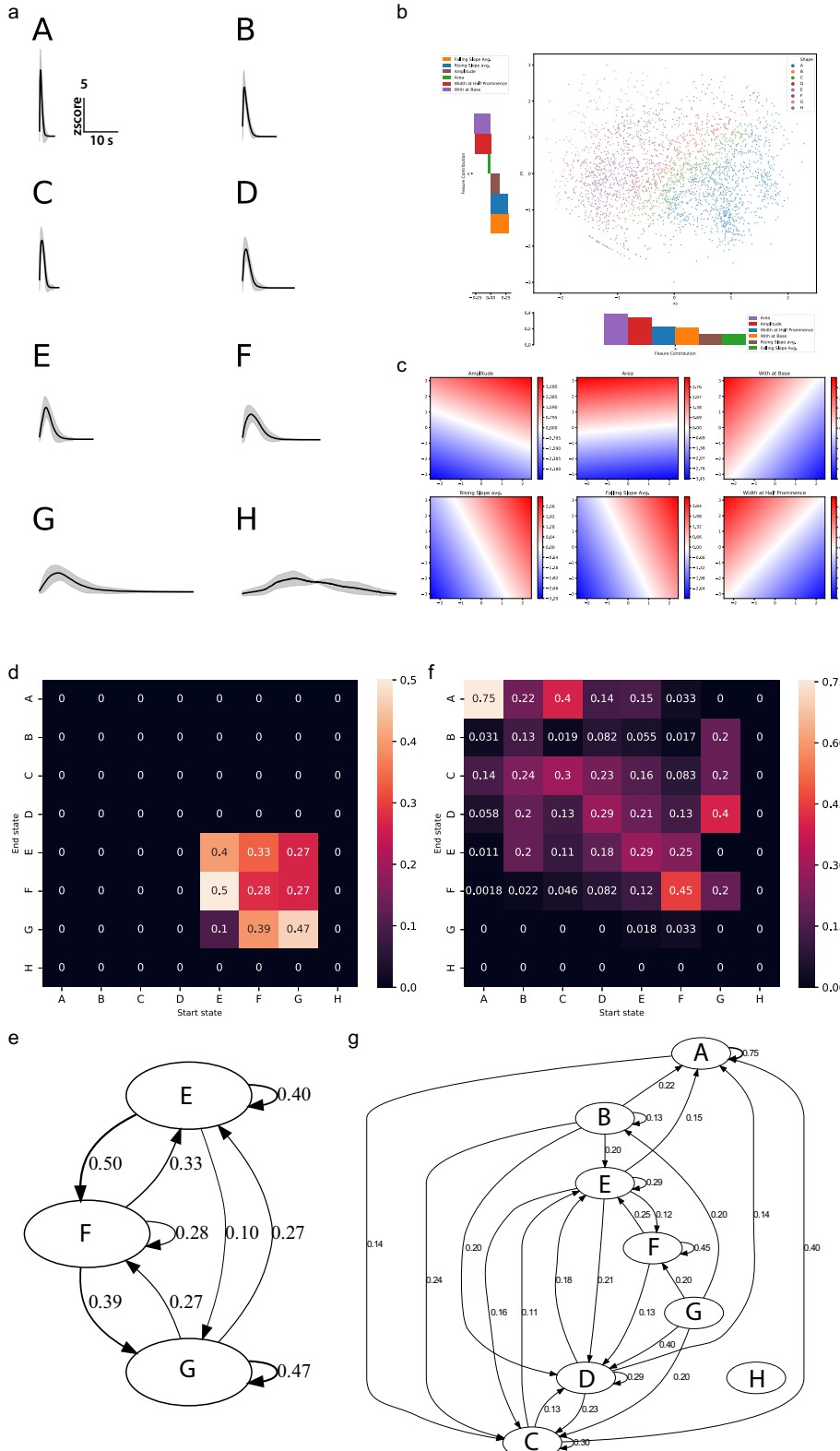

cutting-edge tools such as the deep-learning tool NuSeT, which as we demonstrate can markedly improve the performance of the well-established and popular signal extraction method CNMF(E). Importantly, Mesmerize's capacity to produce FAIR datasets by the encapsulation of raw data, analysis procedures and interactive plots en masse provides a blueprint for other projects and future software platforms. In future directions, Mesmerize could provide

neuroscientists with a user-friendly interface to back-end tools such as DataJoint[6] and NWB[7]. This will help create a community where traceable visualizations and reproducible analysis become more common in the biological sciences.

Mesmerize provides the opportunity to combine functional fingerprinting (calcium signal or other using GEIs) with genetic fingerprinting (e.g. regulatory elements) in genetically tractable

**Fig. 6 k-Shape clustering and Markov chains. a** Cluster means from k-Shape clustering of peaks from neuronal and non-neuronal cells in the head of *C. intestinalis*. Clusters are assigned alphabetical labels according to their half-peak width. Error bands show within-cluster standard deviation. **b** PCA of peak-features showing how k-Shape clustering maps onto the PCA space. **c** Inverse transform for each of the input features showing the characteristics of the PCA space. Color intensity and the color bar scales indicate the magnitude of the corresponding feature. **d** State transition matrix of a Markov chain created from discretized sequences of HNK-1 cell calcium traces and the corresponding (**e**) state transition graph. **f** State transition matrix of a Markov chain created from discretized sequences of photoreceptor cell calcium traces and the corresponding (**g**) state transition graph. Color bar scales in (**d**) and (**f**) are transition probability. Numbers on the transition graphs in (**e**) and (**g**) also show transition probability. Transition probabilities < 0.1 were excluded to reduce visual clutter.

organisms with the potential to simplify systems-level analyses that utilize complex combinations of categorical variables that include multiple genotypes, drugs, and other experimental groups. Our functional imaging analysis of genetically defined neuronal and non-neuronal cell types in *C. intestinalis* showed that different neuronal cell types can be grouped together based on their calcium fingerprint. In addition, it also revealed for the first time some of the basic building blocks that build the observed calcium activity (k-Shape-derived archetypes) and how these building blocks can be organized (Markov chains) in order to generate distinct calcium dynamics. The *C. intestinalis* datasets (both neuronal and non-neuronal) generated in this work will enrich an ever-growing ecosystem of openly available genomic[44,45], morphological and genetic[71–73] resources for an emerging model system for neuroscience and beyond.

## Methods

**Obtaining *C. intestinalis* (type B).** Gravid hermaphrodite adults used in this study were collected from Døsjevika, Bildøy Marina AS near Bergen, 5353, Norway with GPS coordinates: 60.344330, 5.110812.

**Rearing conditions for adult Cionas.** Adult *C. intestinalis* were kept in a purpose-made facility at the Sars Centre. In all, 50–100 adults were housed in 50 L tanks with running sea water with a temperature of 10 °C and pH of approximately 8.2 under constant illumination to enhance egg production[37].

**Electroporation of zygotes and staging of larvae.** Electroporations were performed largely as described by L. Christiaen et al.[74]; adult *C. intestinalis* were dissected to obtain eggs and sperm to perform fertilization in vitro. Eggs were then dechorionated using chemical dechorionation in a pronase with sodium-thioglycolate solution and placed on a rocker for ~6 min until zygotes were fully dechorionated. Dechorionated eggs were washed several times and then fertilized with sperm for ~10 min. After thoroughly washing zygotes were electroporated in a mannitol solution with 70–100 μg of DNA depending on the typical expression levels of a given construct. We electroporated zygotes in MBP Catalog #5540 electroporation cuvettes with a 4 mm gap using a BIORAD GenePulserXcell with a CE-module. The settings we used were Exponential Protocol: 50 V, Capacitance: 800–1000 μF, Resistance ∞ and we aimed for an electroporation time constant of 15–30 ms. Embryos were cultured in ASW (artificial sea water, Red Sea Salt) at 14 °C until they were swimming larvae (Stage26 according to FABA; https://www.bpni.bio.keio.ac.jp/chordate/faba/1.4/top.html) to be used for imaging. From fertilization until we started imaging the average age of the animals was 36 h at 14 °C. We imaged animals that were up to ~44 h post fertilization. For reference, at 14 °C tail regression starts ~52 h post fertilization. The pH of the ASW was 8.4 at 14 °C. The salinity of the ASW was 3.3–3.4%.

**Ciona calcium imaging.** Stage 26 larvae were embedded in 1.5% low melting point agarose (Fisher BioReagents, BP1360-100) between two coverslips to minimize scattering and bathed in artificial sea water. Illumination was provided by a mercury lamp with a BP470/20, FT493, BP505-530 filterset. A Hamamatsu Orca FlashV4 CMOS camera acquired images at 10 Hz with exposure times of 100 ms using a custom application[75] using a python library for interfacing with Hamamatsu cameras[76]. Imaging was performed at 16 °C using a Zeiss Examiner A1 with a water immersion objective ZEISS W B- ACHROPLAN ×40.

**Signal extraction.** Images were motion-corrected using NoRMCorre[22] and signal extraction was performed using CNMFE[24] with parameters optimized per video. Extracted signals that were merely movement or noise were excluded. All parameters for motion correction and CNMFE can be seen in the available dataset. Cells were identified with the assistance of the connectome[41,42] to the best of our capability with 1-photon data (Supplementary Fig. 3). Only regions that covered cell bodies were tagged; axons were not tagged with cell identity labels.

**Hierarchical clustering.** Analysis was performed using the Mesmerize flowchart. All traces extracted from CNMFE were normalized between 0 and 1. The DFT of the normalized data was calculated using 'scipy.fftpack.rfft' from the SciPy (v1.3) Python library[28]. The logarithm of the absolute value of the DFT data arrays was taken, and the first 1000 frequency domains (corresponding to frequencies between 0 and 1.67 Hz) were used for clustering. This cutoff was determined by looking at the sum of squared differences (SOSD) between the raw curves and interpolated inverse Fourier transforms (IFTs) of the DFTs with a step-wise increase in the frequency cutoff (Supplementary Fig. 4). The SOSD changes negligibly beyond 1.67 Hz, and inclusion of higher frequencies would likely introduce noise. At 1001 frequency domains, corresponding to 1.676 Hz, the cumulative sum of the mean SOSD corresponds to 94.5% of the total cumulative sum from all frequency domains (i.e. all domains up to Nyquist frequency). EMD was used as the distance metric through the OpenCV[77] (v3.4) EMD function and complete linkage was used for constructing the tree. The dendrogram was cut to obtain four clusters according to the maxima of the silhouette scores (Fig. 4k). The Davies−Bouldin score was also relatively low for the four clusters (Fig. 4l). Silhouette scores were calculated using sklearn[29] v0.23 and a custom-written function was used to adapt the Davies−Bouldin score for EMD. Euclidean Davies Bouldin scores were calculated using sklearn[29] v0.23.

**k-Shape clustering.** This method uses a normalized cross-correlation function to derive a shape-based distance metric[65]. The tslearn[30] implementation is used in Mesmerize. Tslearn v0.4 was used. Peak-curves were used as the input data for k-Shape clustering and the parameters can be seen in Supplementary Fig. 5. A gridsearch was performed to optimize the hyperparameters and obtain a set of clusters with minimum inertia (sum of within-cluster distances) with no empty clusters. The search range for the number of clusters to form was 2–14. For each iteration of the gridsearch, peak-curves were ordered based on half-peak-width and partitioned into n_cluster partitions and a random centroid seed was picked from each partition.

**Markov chains.** Cluster membership of peaks, as determined through k-Shape clustering, was used to express calcium traces as discretized sequences. These sequences were used to create Markov chain models using the pomegranate[78] Python library.

**Determining stimulus tuning of cell within the CRCNS pvc-7 and zebrafish datasets.** All stimulus periods were extracted and the average response was calculated for each stimulus, such as an orientation, spatial frequency, or temporal frequency for the pvc-7 dataset; or heat-on, heat-off, and none (inter-trial period). The stimulus tuning of the cell was then determined as the stimulus that produced the highest mean response in that cell. For more details, this is calculated by the 'get_tuning_curves()' function within 'mesmerize.plotting.widgets.stimulus_tuning.widget'. The analysis graph for the analysis of the pvc-7 dataset can be seen in Supplementary Fig. 1, and the analysis graph for the analysis of the zebrafish dataset can be seen in Supplementary Fig. 6.

**Linear discriminant analysis.** The Neural Decompose node was used in the Mesmerize flowchart to perform supervised LDA. Each timepoint of the recording is used as a feature vector containing the intensity values for each cell at that timepoint. The model was trained using the stimulus periods (heat-on, heat-on-delayed, and none) for classification.

**Promoters.** To drive the expression of GCaMP6s population in different cell types in *C. intestinalis* larvae, we used eight different promoters. Details are shown in Supplementary Table 4. Sequences for several of these promoters were obtained from DBTGR[73]. To amplify these promoters *C. intestinalis* gDNA, which was purified using the Wizard Genomic DNA Purification Kit (A1120, Promega). Using purified gDNA at 150 ng/μl, the primers shown in Supplementary Table 5 dNTPs (Thermofisher, R0182) and the Q5 High-Fidelity DNA Polymerase (M0491L, NEB) we performed PCR reactions. The amplified PCR products were gel purified using Zymogclean Gel DNA Recovery Kit (Zymo research, D4002) and inserted into P4-P1R vector using BP Clonase II (Invitrogen, P/N56480). Positive clones identified by restriction digest were sequenced. Subsequently, we performed a four-way Gateway Recombination using one of the promoters in the first position, GCaMP6s in the second position and unc-54 3′ UTR in the third position. These were recombined into a pDEST II backbone using LR Clonase II (Invitrogen, P/N56485). Expression constructs were electroporated at a range of concentrations (80–120 μg).

**Statistics and reproducibility**. The details on the number of animals and trials per *C. intestinalis* promoter imaged are indicated in Supplementary Table 1. Each GCaMP6s construct was electroporated at least two times and larvae from two or more independent electroporations were imaged. All biological replicates were included in our analysis. CNMFE extracted signals that represented movement in the FOV or noise were excluded. Signals from heavily out of focus regions or cells were also excluded. *C. intestinalis* micrographs in Fig. 1a and Supplementary Fig. 3 are representative maximum projections from PC2 > GCaMP6s larvae single movies each of which composed of 3000 frames. For the zebrafish micrographs in Figs. 1a and 3 are representative maximum projections of a single plane from brain stacks that each contained 30 planes (each imaging plane was probed with three stimulus trials). For mouse brain micrographs are maximum projections from individual movies containing >20,000 frames.

**Reporting summary**. Further information on research design is available in the Nature Research Reporting Summary linked to this article.

## Data availability

The imaging datasets generated are available as a Mesmerize project and can be downloaded from Figshare: *C. intestinalis*: https://doi.org/10.6084/m9.figshare.10289162 [79]; The CRCNS pvc-7 dataset used in this study is provided as a Mesmerize dataset: https://doi.org/10.6084/m9.figshare.10293041 [80]. The Zebrafish dataset used in this study is provided as a Mesmerize dataset here: https://doi.org/10.6084/m9.figshare.14748915 [81].

## Code availability

The code for Mesmerize has been deposited in the following Github repo: https://github.com/kushalkolar/MESmerize. The Mesmerize GitHub repo with the code has been archived in Zenodo: https://doi.org/10.5281/zenodo.5539440 [82]. GitHub repo for Mesmerize: https://github.com/kushalkolar/MESmerize. Notebooks that produce some of the figures are available on GitHub: https://github.com/kushalkolar/mesmerize_manuscript_notebooks. Many of these notebooks can be run on MyBinder: https://mybinder.org/v2/gh/kushalkolar/mesmerize_manuscript_notebooks/master. Mesmerize can be installed through pip on all platforms: https://pypi.org/project/mesmerize/. We provide a ready-to-use VM with Mesmerize and all features pre-installed. You can run this VM on Windows, Mac OSX, or Linux. Please visit: http://docs.mesmerizelab.org/en/master/user_guides/installation.html#all-platforms. Thorough Mesmerize documentation can be found here: http://docs.mesmerizelab.org/. Gitter community for discussion: https://gitter.im/mesmerize_discussion/community. Video tutorials: https://www.youtube.com/playlist?list=PLgofWiw2s4REPxH8bx8wZo_6ca435OKqg. Additional video tutorials: https://www.youtube.com/playlist?list=PLgofWiw2s4RF_RkGRUfflcj5k5KUTG3o.

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

## Acknowledgements

We would like to thank Mesmerize users and the community for their engagement in the gitter channel and GitHub for constant feedback and bug reports. We would like to thank Pietro Vertechi and Julius Parulek for technical advice and members of the Chatzigeorgiou lab for user feedback during Mesmerize's early development. We thank Mie Wong and Dario Sarra for comments on the manuscript. This project has been funded by a grant of the Research Council of Norway, of which M.C. is the PI: grant number 234817 (Sars International Centre for Marine Molecular Biology Research, 2013-2022).

## Author contributions

K.K. and M.C. conceived, supervised, and directed the project. K.K. wrote Mesmerize and analyzed all experiments. D.D. aided and contributed to the development of Mesmerize and provided critical input. Imaging experiments were performed by K.K. and M.C. GCaMP6s constructs were cloned by M.C., J.C.Z. and J.H. assisted with significant user testing of the Mesmerize platform and aided in development. J.C.Z. created the Mesmerize logo. The manuscript was written by K.K. and M.C. All authors commented on the manuscript.

## Competing interests

The authors declare no competing interests.
