## [Peer Review File · Nature Communications]

Mesmerize is a dynamically adaptable user-friendly analysis platform for 2D and 3D calcium imaging data.Reviewers' Comments:

Reviewer #1:

Remarks to the Author:

Data handling is a major challenge in today's biomedical research. New technology developments can produce large quantities of data that is difficult to analyse and extract relevant information from. Therefore, new software tools as described in this manuscript are required to maximize the output of these new methods. Recently, new software packages have been reported, for example those mentioned in this manuscript, CaImAn (Giovannucci, A. et al. CaImAn an open source tool for scalable calcium imaging data analysis. *Elife* (2019) doi:10.7554/eLife.38173.), Suite2p (Pachitariu, M. et al. Suite2p: beyond 10,000 neurons with standard two-photon microscopy. *bioRxiv* (2016) doi:10.1101/061507.) and SIMA (Kaifosh, P. et al. SIMA: Python software for analysis of dynamic fluorescence imaging data. *Front. Neuroinform.* (2014) doi:10.3389/fninf.2.014.00080.).

In this study, Kolar et al. presents Mesmerize: A versatile platform for calcium imaging analysis and creation of self-contained FAIR datasets. To demonstrate the applicability of their platform they analyse one dataset of recordings from the mouse visual cortex and one dataset of recordings from *C. elegans* motor neurons.

From a technology viewpoint, this study is no more than the combination of previously established techniques packed in a new graphical interface. From a biological standpoint, Mesmerize may be potentially important in basic research for characterizing calcium signalling in various tissues. However, authors failed to demonstrate what can be newly investigated with their software tool. Without significant advance and broad impacts, I believe that this work better fits a more specialized journal on neuroscience methods.

Major points:

My main concern with this work is that it shows no biological insight. It is true that there is a need for software algorithms and tools that can analyse large amounts of recorded data. But in order for the analysis to be useful, the authors must show that Mesmerize can be used to analyse medical/biological issues. For example, the authors can use Mesmerize to analyse spontaneous calcium signalling in primary visual cortex before and after eye opening. In this way Mesmerize should be able to pick up differences in patterns of calcium activity. It must also be shown that the analysis can be repeated with similar results that can be used to perform significant tests.

The manuscript does not mention that Mesmerize can be used to analyse three-dimensional (3D) calcium imaging recordings. This is now becoming more and more important with the development of advanced light microscope techniques, for example light-sheet microscopy. With light-sheet microscopy, whole *C. elegans* and zebrafish can be studied with high temporal and spatial accuracy. 3D data handling with Mesmerize would make the platform more unique and useful for the biomedical research field.

A richer discussion on how to interpret the analysis data from Mesmerize from a biological point of view is needed. What does it mean that two cells and their calcium traces are clustered? If they are close to each other, it is likely that the cells are connected in some way. But if they are far apart, what does it mean to be clustered? Please elaborate.

Minor points:

Abstract, the name Mesmerize could be introduced better.

Introduction, Fig 1b is mentioned before Fig 1a.

Results, Fig 2a is not mentioned.

Images, scale bars are missing.

Reviewer #2:
Remarks to the Author:
Summary

The authors present an graphical user interface integrated with existing algorithms to pre- and post-process datasets collected with calcium imaging. The interface provides routines to organize data, to define analysis pipelines, to visualize and interact with the data, to perform analyses on the neuronal population traces, and to generate reproducible e-figures.

Major issues

* The paper is very difficult to parse. It would benefit from reorganization and clear separation between methods, results and contribution versus the state-of-the-art. For each of these sections, subsections should be used to list each contribution. Figures are sometimes small and not very informative.

* Although a very laudable effort, the software does not seem to be ready for prime time. The installation was super smooth both in Linux and Mac OS. I tried to use the software and I was able to load and visualize a movie, but not able to add anything to the batch manager to start motion correcting. I could therefore not test any other functionality downstream. I tried to consult the documentation but there was very little information about how to create such batch (or at least I was not able to find it). I played for about 15 minutes with it and then I stopped. I think this could cause frustration in users. There should be a way to perform interactively operations without needing to resort to the batch manager. Maybe there is but I could not find it.

* It is not clear what the main contribution of the paper is. What is the relationship with the state of the art? Why is this framework better than other existing platforms? What new problem is it solving? Is this paper making any specific claim about biological findings (A)? Is this manuscript introducing new computational methodologies for the analysis of neuronal population in calcium imaging data (B)? Is this paper introducing a generalized and flexible platform for the analysis of calcium imaging data (C)? In each of the cases above many questions need to be answered and comparison with the state of the art must be tackled.

A) The author seems to hint at some biological findings, but they are not clearly delineated. If this is a part of the claims included in the paper it should better developed and organized. The results section should be decoupled from the methods and clear biological claims should be made, aligned with data clearly supporting the conclusions.

B) In this case the authors should also relate to the state-of-the-art. It is the reviewer's opinion that the authors are wrapping existing tools. Is the application of these tools novel for calcium imaging? For instance, the authors report "When used for hierarchical clustering, the EMD of frequency domain representations leads to better separation of disparate dynamics and an aggregation of similar dynamics (Fig 3e-g).", however, this needs to be supported with quantitative statements and related to the state of the art.

* <https://bmcbiol.biomedcentral.com/articles/10.1186/s12915-018-0606-4>

* and others

C) How to deal with multiple data formats and ontologies (especially in connection with behavioral data)? Is this platform been employed already by other users besides the developer and researchers within the same laboratory? What form of support is provided to users? How easy is to integrate new

steps into the analysis pipeline? How does this framework compare with existing tools?

* <https://journals.plos.org/ploscompbiol/article?rev=1&id=10.1371/journal.pcbi.1005526>

* <https://www.frontiersin.org/articles/10.3389/fninf.2017.00044/full>

* And more ...

This seems a very well designed application to solve the analysis problems tackled in a laboratory. The manuscript would benefit from a clear pathway to have these methods generalize across laboratories and scientific questions.

Minor comments:

* FAIR: abbreviation not introduced

* TLC: abbreviation not introduced

* Many figure panels are very difficult to read. The fonts and graphics are often too small. Examples: Fig 1 b, h, i, j, d ... etc.

* "Simple data structures, outlined in the API, allow the imaging data to potentially originate from any model organism" -> not clear what does this refer to, paper should be self contained

* Reference to Fig 1b appears before reference to Fig 1a

Reviewer #3:

Remarks to the Author:

Calcium imaging is a powerful tool for analyzing cellular activity of nervous systems, embryos, and other various tissues. The authors developed a new calcium imaging analysis platform, Mesmerize, with which users can perform the whole analysis process from raw data to analyzed data presentation. The platform is flexible and expandable. The authors demonstrate usefulness of this new platform by using GCaMP6 calcium imaging data of whole larvae of *Ciona intestinalis* and also using mammalian brain and *C. elegans* data sets. The authors' work also demonstrates the power of this emerging model chordate to study development and function of the nervous systems and some other tissues. I think this work has a substantial impact for various fields of biology and life sciences, including neurobiology, physiology and developmental biology, and therefore, I support publication of this manuscript in *Nature communications* if the following problems are properly solved.

Most of my concerns are about organization of the manuscript and insufficient presentation of biological materials and experimental methods as summarized below.

The very long Introduction seems to contain substantial amount of detailed explanation of the Mesmerize platform. This is somewhat confusing. I think moving these parts to Results & Discussion will make the manuscript easier for readers to understand.

Some essential information on biological materials seems to be lacking. There is no description of biological procedures using the main experimental organism (*Ciona*), such as how the authors obtained the animals and embryos, how they transfect GCaMP6 constructs into embryos, how they reared the embryos and larvae, and what temperature was used during development and imaging analysis of embryos and larvae.

Ciona is a marine invertebrate chordate and genetically diverse between different populations. Genetically homogeneous strains have not been established. I assume that the authors used *Ciona intestinalis* (type B), the species whose connectome has been solved (Ryan et al., 2016, 2019). However, many other key studies, including single cell transcriptomic studies (Sharma et al. 2019; Cao et al., 2019) and the genome project (Dehal et al., 2002; Satou et al., 2019) came from the related species *Ciona robusta* (*Ciona intestinalis* type A). Therefore, it is important to precisely

describe the geographic origin or population of *Ciona* used in the study.

Information on detailed developmental stages of *Ciona* larvae used in this study is also important because modes of larval behavior change during larval stages and thus neural circuits and neuronal functions may change during larval development (Nakagawa et al., *Photochem. Photobiol.* 70, 359–362, 1999; Tsuda et al., *J. Exp. Biol.* 206, 1409–1417, 2003; Tsuda et al., *Zool. Sci.* 20, 13–22, 2003). Therefore, the developmental stages of the larvae observed in this study should be described, for example, by showing time (hours) after fertilization along with incubation temperature and/or stages according to the standard developmental table of *Ciona* (FABA/FABA2; <https://www.bpni.bio.keio.ac.jp/chordate/faba/1.4/top.html>).

Experimental procedures for the imaging of *Ciona* larvae should be described in some more detail. The authors just mentioned that larvae were embedded in low melting point agarose. Concentration (%) of low melting point agarose should be described. Also, information on the microscope used should be mentioned.

Cell-type identification presented in this manuscript is not very convincing. In the Signal Extraction subsection of Methods, the authors mention that cells were identified with the assistance of the connectome to the best of their capability with 1-photon data. However, as shown in Table 1, most neural promoters used label many cell types. How did they identify the cells? The connectome was obtained by using a larva developed with an egg envelop (chorion) (Ryan et al. 2016). However, reporter gene transfection into *Ciona* embryo is usually performed by electroporation with dechorionated eggs. It is known that dechoriation disturbs left-right asymmetry of the larval brain (Shimeld and Levin, 2006; Oonuma et al. 2016). This would make the comparison with connectome data difficult. Representative examples of identified cell type images should be presented.

Minor points:

In the first paragraph of Results & Discussion, there are some mis-citations of Figure 2. Fig 2b, Fig 2c, Fig 2d, and Fig 2e may be corrected to Fig 2a, Fig 2b, Fig 2c, and Fig 2d, respectively.

*Reviewer #1, an expert in calcium signalling, with some expertise in cluster analysis
(Remarks to the Author):*

*Data handling is a major challenge in today's biomedical research. New technology developments can produce large quantities of data that is difficult to analyse and extract relevant information from. Therefore, new software tools as described in this manuscript are required to maximize the output of these new methods. Recently, new software packages have been reported, for example those mentioned in this manuscript, CaImAn (Giovannucci, A. et al. CaImAn an open source tool for scalable calcium imaging data analysis. *Elife* (2019) doi:10.7554/eLife.38173.), Suite2p (Pachitariu, M. et al. Suite2p: beyond 10,000 neurons with standard two-photon microscopy. *bioRxiv* (2016) doi:10.1101/061507.) and SIMA (Kaifosh, P. et al. SIMA: Python software for analysis of dynamic fluorescence imaging data. *Front. Neuroinform.* (2014) doi:10.3389/fninf.2014.00080.).*

*In this study, Kolar et al. presents Mesmerize: A versatile platform for calcium imaging analysis and creation of self-contained FAIR datasets. To demonstrate the applicability of their platform they analyse one dataset of recordings from the mouse visual cortex and one dataset of recordings from *C. elegans* motor neurons.*

From a technology viewpoint, this study is no more than the combination of previously established techniques packed in a new graphical interface. From a biological standpoint, Mesmerize may be potentially important in basic research for characterizing calcium signalling in various tissues. However, authors failed to demonstrate what can be newly investigated with their software tool. Without significant advance and broad impacts, I believe that this work better fits a more specialized journal on neuroscience methods.

Major points:

1. My main concern with this work is that it shows no biological insight. It is true that there is a need for software algorithms and tools that can analyse large amounts of recorded data. But in order for the analysis to be useful, the authors must show that Mesmerize can be used to analyse medical/biological issues. For example, the authors can use Mesmerize to analyse spontaneous calcium signalling in primary visual cortex before and after eye opening. In this way Mesmerize should be able to pick up differences in patterns of calcium activity. It must also be shown that the analysis can be repeated with similar results that can be used to perform significant tests.

We respect the opinion of the reviewer and maybe it was our shortfall not to convey as required the important technical and biological advances we have made. We believe that this work is of high significance and novelty within the Ciona and broader protochordate field (as stated by reviewer 3). Although state-of-the-art neuronal studies in a number of established model organisms (e.g. mice, drosophila, *C. elegans* and zebrafish) are much more sophisticated, no study has previously characterized in a quantitative manner spontaneous activity across such a plethora of neurons in *C. intestinalis* larvae, which is an emerging model organism for systems-level neuroscience. The results and data from this study are available as a FAIR dataset, which provides the ascidian (including *C. intestinalis*) and the wider invertebrate chordate community with a valuable reference resource for future neurobiological studies. We would like to note that the ascidian community has a strong commitment to FAIR data as indicated for example by a recent publication on the ANISEED

database which is the ascidian equivalent to Wormbase and FlyBase (Dardaillon et al., NAR 2019 <https://doi.org/10.1093/nar/gkz955>).

In addition, we believe that two analytical tools that we incorporated into Mesmerize will be of substantial importance for the analysis of calcium activity patterns not only in the context of neurons but also in non-neuronal tissues across a wide range of experimental contexts and animal models. Firstly, the development of the Discrete Fourier Transform (DFT) – Earth Mover’s Distance (EMD) clustering approach presents a novel step forwards in the analysis of spontaneous activity/noise in different neurobiological process such as development, encoding and stochastic resonance. In our revised manuscript we show that DFT-EMD based clustering outperforms more commonly used methods for clustering neuronal activity (please see Figure 4i-l, lines 366-377). Importantly, since our approach is suitable for data that are not temporally aligned it has significant potential for the analysis of spontaneous activity during circadian cycles, development (of both neuronal & non neuronal tissues) and pathological conditions using as psychiatric disease models.

The second technical advance we make through Mesmerize is the ability to provide a quantitative method to define discrete archetypical shapes from calcium traces (akin to a calcium signaling alphabet) which has been a long-standing requirement in the field, especially in the context of complex and irregular calcium activity observed during embryonic development or in pathological conditions (e.g. seizures). To date such archetypes were either defined in a subjective manner (as ‘puffs’, ‘blips’, etc) or by grouping peaks according to features like amplitude, width, slope etc. In order to introduce a contemporary method for defining the archetypes, or motifs, within calcium imaging traces we incorporated k-Shape clustering. This machine learning approach is provided by a few libraries, such as tslearn. These state of the art tools are now much more approachable by a broad range of biologists through the Mesmerize platform, which allows users to perform peak detection (including manual curation if required) and then perform a hyperparameter gridsearch to obtain an appropriate model (i.e. set of archetypes) for their data. (lines 437-464, please see Figure 6). Furthermore, biologists can use k-Shape derived archetypes in combination with statistical models, such as Markov Chains, to understand similarities and differences in the structure and organization of calcium dynamics across different cell types (for example please see Fig.6f-g).

2. The manuscript does not mention that Mesmerize can be used to analyse three-dimensional (3D) calcium imaging recordings. This is now becoming more and more important with the development of advanced light microscope techniques, for example light-sheet microscopy. With light-sheet microscopy, whole c elegance and zebrafish can be studied with high temporal and spatial accuracy. 3D data handling with Mesmerize would make the platform more unique and useful for the biomedical research field.

We thank the reviewer for raising this extremely important point. We agree with them and to this aim, Mesmerize is now able to handle 3D volumetric imaging datasets with the same annotation and analysis capabilities that we provide for 2D datasets. To

demonstrate this new functionality of Mesmerize we analyzed an *in-vivo* 2-photon imaging dataset where zebrafish larvae expressing a nuclear localized GCaMP are presented with various forms of thermal stimuli (Haesemeyer et al., Neuron 2018) (please see Figure3, lines277-301).

We encourage the reviewer to view the Mesmerize 3D Tutorial which walks the user through using Mesmerize for 3D calcium imaging data:

https://www.youtube.com/watch?v=0U3Zn7SqKUc&list=PLgofWiw2s4RF_RkGRUf_flcj5k5KUTG3o_&index=5

A challenge that we faced with this dataset was the relatively poor performance of the standard ROI/signal extraction packages (CNMF) possibly due to the lower signal-to-noise ratio, which is a common issue with 2-photon volumetric imaging (Keller & Ahrens, Neuron, 2015). To improve the ROI extraction, we took advantage of a deep-learning tool for cellular segmentation called NuSeT (Yang et al., PLOS Comput. Biol., 2020) that we have incorporated as a module within Mesmerize. We processed the zebrafish brain dataset first with NuSeT to generate a binary mask which was then used to initialize CNMF. This combinatorial approach markedly improved ROI identification and signal extraction (please see Figure3b). Since the NuSeT network is trained for nuclear data, the incorporation of NuSeT and NuSeT + CNMF within Mesmerize would be extremely useful for anyone using a nuclear GCaMP. Furthermore, this illustrates a key attribute of Mesmerize, the ability to combine multiple cutting-edge tools, allowing for higher quality analysis that can be performed non-computational biologists.

While a detailed re-analysis of the Haesemeyer et al paper is beyond the scope of our work, we generated tuning curves for all identified ROIs/neurons and we use these to train a Linear Discriminant Analysis classifier to identify three distinct brain states in response to sensory stimulation.

3. A richer discussion on how to interpret the analysis data from Mesmerize from a biological point of view is needed. What does it mean that two cells and their calcium traces are clustered? If they are close to each other, it is likely that the cells are connected in some way. But if they are far apart, what does it mean to be clustered? Please elaborate.

We agree with reviewer that we should have elaborated on the biological significance/ interpretation of our Ciona dataset. At the same time, we felt that we had to elaborate on the Discrete Fourier Transform (DFT) – Earth Mover’s Distance (EMD) clustering approach, which is central to the analysis we performed. To these aims:

1. We have explained DFT-EMD in more detail so that it becomes conceptually more accessible to the readers (lines 334-377) and we provide a more thorough visual illustration of our approach in Figure 4.
2. We have now elaborated on how DFT-EMD separates effectively epidermal and mesenchymal cells from neurons based on their radically different calcium fingerprints (a situation reflecting cells that are far apart in our clustering analysis) (lines 382-397). Additionally, we discuss what the role of these dynamics may be in terms of cellular functions (lines 417-421).

3. We have elaborated on our interpretation of the neuronal clustering analysis. We suggest that our findings support previously published anatomical and molecular studies implicating: a) four cell types with a strong glutamatergic signature (PSNs, RTENs, aATENs and pATENs) in the same somatosensory circuit (lines 402-408) ; b) five classes of interneurons with high activity some of which have been postulated based on their numerous synaptic partners to act as ‘hubs’ in the larval nervous system (lines 409-417).

4. From a broader biological perspective we have now highlighted the potential value of using DFT-EMD in quantifying spontaneous calcium dynamics and neuronal noise across different contexts (e.g. development, encoding, stochastic resonance, and possibly others) (lines 422-435).

Minor points:

Abstract, the name Mesmerize could be introduced better.

We have changed the abstract to read as follows (lines 29-32): “Here we present “Mesmerize”, an efficient, expandable and user-friendly analysis platform, which uses a Findable, Accessible, Interoperable and Reproducible (FAIR) system to encapsulate the entire analysis process, from raw data to interactive visualizations for publication. ” We hope that this new sentence does a better job at introducing both Mesmerize and the FAIR principles which we consider to be a key aspect of Mesmerize.

Introduction, Fig 1b is mentioned before Fig 1a.

We thank the reviewer for pointing this out. We now mention Fig 1a before Fig 1b. In the Rich Data Annotation section (lines 131-133) it now reads as follows: “...zebrafish, and *Ciona intestinalis* (Fig 1a). These datasets can be visualized using the Mesmerize Viewer, which provides GUI front-ends (based on pyqtgraph) and API interfaces for various signal extraction modules (Fig 1b).”

Results, Fig 2a is not mentioned.

We thank the reviewer for pointing out this omission. This has now been corrected. Please see the beginning of Usage Examples section (lines 254-255): “ ..., which consists of *in-vivo* 2-photon imaging data from layer 4 cells in the mouse visual cortex³⁵ (Fig 2a).”

Images, scale bars are missing.

We apologize for this omission. We have now included scale bars in Figures 1a (*Ciona*), Figure 3 b,f (zebrafish), Supplemental Figure 1 (*C.elegans*). Unfortunately, the pvc-7 mouse dataset did not provide the relevant pixel/ μm conversion factor.

Reviewer #2, an expert in calcium imaging algorithms (Remarks to the Author):

Summary

The authors present an graphical user interface integrated with existing algorithms to pre- and post-process datasets collected with calcium imaging. The interface provides routines to organize data, to define analysis pipelines, to visualize and interact with the data, to perform analyses on the neuronal population traces, and to generate reproducible e-figures.

Major issues

** The paper is very difficult to parse. It would benefit from reorganization and clear separation between methods, results and contribution versus the state-of-the-art. For each of these sections, subsections should be used to list each contribution. Figures are sometimes small and not very informative.*

We thank the reviewer for these suggestions. In the revised manuscript we have more clearly organized the description of the various part of the Mesmerize platform and provided comparisons with other state-of-the-art tools. We have also tried to make the figures more informative where possible.

In particular, the revised introduction provides a detailed comparison of Mesmerize with other state-of-the-art tools in both calcium imaging analysis and management of imaging data (lines 40-126). To provide a visual summary of these comparisons we have generated Table 1. The next section of the revised manuscript titled ‘Mesmerize Platform’ focuses on key methodological and technical aspects of Mesmerize (lines 127-248). The third section of the revised manuscript is titled ‘Results’ concentrating primarily on the biological results we obtained using the Mesmerize platform. In accordance with the reviewer’s suggestions, we discuss the methodological aspects and the results in several subsections. While we have tried to reduce the overlap between results and methods, we believe that certain technical aspects of Mesmerize namely stimulus mapping, volumetric calcium imaging data handling and analysis, novel clustering analysis using Earth Mover’s Distances (EMD) between discrete Fourier transforms (DFT), and k-Shape analyses methods do warrant discussion in combination with the biological findings we present. We feel that our Ciona results act as interesting illustrations of how these technical developments can be applied to biological questions, and the Mouse & Zebrafish sections provide straight forward examples of more common neurobiological analysis in order to introduce new users to the Mesmerize platform’s structure..

Regarding the manuscript’s figures, we have drastically modified Figure 1, providing what we think is a better and more informative overview of the Mesmerize platform and its capabilities. For example, just by looking at the revised panel a, the reader can appreciate that Mesmerize can handle, 1-photon, 2-photon and volumetric calcium imaging datasets, while panel b clearly illustrates the modular nature of Mesmerize by highlighting some of the integrated modules and available data importers. Figure 2, which illustrates the analysis of mouse visual cortex calcium dynamics in response to visual sinusoidal grating stimuli has been updated by adding what we consider to be an informative overview of the steps required in Mesmerize to perform the analysis of the data. In this way the readers can appreciate the relative simplicity of the analysis

process in Mesmerize. We acknowledge that the DFT-EMD clustering and k-Shape analysis merited clearer explanations and more informative figures so we made a dedicated figure for DFT-EMD clustering (please see Figure 4) and k-Shape analysis (please see Figure 6). In an effort to present our most important biological findings with as clear separation from the methods as possible we generated a separate figure on imaging and analysis of spontaneous calcium dynamics in Ciona neuronal and non-neuronal populations (please see Figure 5).

** Although a very laudable effort, the software does not seem to be ready for prime time. The installation was super smooth both in Linux and Mac OS. I tried to use the software and I was able to load and visualize a movie, but not able to add anything to the batch manager to start motion correcting. I could therefore not test any other functionality downstream. I tried to consult the documentation but there was very little information about how to create such batch (or at least I was not able to find it). I played for about 15 minutes with it and then I stopped. I think this could cause frustration in users. There should be a way to perform interactive operations without needing to resort to the batch manager. Maybe there is but I could not find it.*

We thank the reviewer for the positive comment regarding the installation in Linux and Mac OS. The current version of Mesmerize 0.7.1 is also available for Windows. We think this is an important development since it increases the potential pool of users. Furthermore, we provide a ready-to-use VM that can be imported in VirtualBox which comes with Mesmerize pre-installed. This allows anyone on most operating systems to get Mesmerize and start working on their data within minutes.

We apologize to the reviewer for the difficulties they faced while trying to use the batch manager. In order to make Mesmerize easier to learn, we have recorded 2.5 hours of detailed tutorials spread over 13 videos. These tutorials cover the entire platform, taking the user from raw data to interactive visualizations by the end of the tutorial series.

We encourage the reviewer to please visit:

https://www.youtube.com/playlist?list=PLgofWiw2s4REPxH8bx8wZo_6ca435OKqg

An additional video series provides additional tutorials, such as using the API:

https://www.youtube.com/watch?v=a1UO2-OhIHw&list=PLgofWiw2s4RF_RkGRUfflcj5k5KUTG3o_&index=3

We respond relatively quickly to issues posted on the GitHub page and on gitter (which are quite active):

<https://github.com/kushalkolar/MESmerize/issues?q=is%3Aissue+is%3Aclosed>
https://gitter.im/mesmerize_discussion/community

Based on the feedback we have received from external Mesmerize users, the combination of the gitter discussion room & the video tutorials make Mesmerize a relatively easy to adopt and use software.

We have expanded upon the existing documentation and we now provide 200 page of detailed detailed documentation as supplemental material to this manuscript.

The Mesmerize Batch Manager performs all computationally intense operations in the Viewer. We opted for this approach since it serves the purpose of aiding users with time-consuming steps such as parameter optimization. For example, parameter optimization of various Caiman tools (such as NoRMCorr motion correction or CNMF(E)) can become difficult to track within jupyter notebooks, especially for non-computational biologists and novice users. The Batch Manager allows both computational & non-computational users to save extremely large amounts of time since they can create a batch with various parameter variants for each of their recordings, allow the batch to run, and come back to see which parameter variants worked for their videos. Users can simply click through a list in the Batch Manager to view the output & parameters for each of their parameter variants. Simple “green” and “red” background labels are used for the items on the Batch Manager list as a user friendly way to check progress and whether there were issues with certain batch items.

Creating a new batch and adding a batch item is straight forward and we are sorry that it was not clearly explained in the previous version of the documentation. For batch creation, the user just selects a location for creating a new batch folder, and then they click the “Add to batch” button on the respective module (such as caiman motion correction, CNMF(E), etc.). We think that the video tutorials will allow new users to come to Mesmerize more easily.

For batch creation specifically please see the following tutorial from 13:15 onward: <https://www.youtube.com/watch?v=D9zKhFkcKTK&t=795s>

** It is not clear what the main contribution of the paper is.*

We would like to apologize if we were not clear enough in our discussion of the major contributions of the paper to the field. To address the reviewer’s comments in the revised manuscript we have made several additions and modifications to the text and the figures.

What is the relationship with the state of the art? Why is this framework better than other existing platforms?

We now compare Mesmerize to the state-of-the-art in terms tools/platforms for calcium imaging analysis and imaging data management (please Table 1 and the introduction). Compared to all other calcium imaging analysis tools, various annotation tools for ROI, stimulus, and sample (categorical labels for movies) are provided to simplify and accelerate the analysis process. These annotations are carried through downstream analysis which allows users to perform some of the most common types of analysis, such as looking at relationships between calcium activity and stimulus/behavior, clustering analysis, and more contemporary forms of analysis which we used to analyze our Ciona dataset. Finally, all of this is seamlessly packaged as a FAIR dataset which the user can easily share with other scientists. None of the current calcium imaging analysis tools provide these systems for organization, downstream analysis, and visualization. More generalize biomedical image analysis tools, such as OMERO, provide some annotation, organization, and basic visualization tools but they are not suitable for calcium imaging data and not as extensive and

flexible as Mesmerize – which supports unlimited categories for ROI, Temporal, and Sample annotations.

What new problem is it solving?

The platform is trying to solve the problem of data organization, sharing, reproducibility and accessibility of cutting-edge analysis to non-computational biologists. To demonstrate how Mesmerize can tackle these issues while also performing novel biology, we have introduced novel and contemporary downstream analysis tools for calcium imaging analysis and show how they can be used to analyze spontaneous calcium activity in neuronal and non-neuronal cells in *Ciona intestinalis*. We have tried to elaborate and more clearly illustrate these challenges in the beginning of our introduction and biological discussion on *Ciona*.

In addition to addressing the above challenges, from a biological point of view, we have tried to the best of our abilities to highlight and clarify some of the key problems/gaps in the field that Mesmerize is solving. Briefly, our novel approach for clustering calcium dynamics time-series using the EMD between DFTs is a new tool which address the problem of quantifying spontaneous calcium dynamics across neurobiological and developmental contexts. It works well with data where the activity are not temporally aligned in contrast to other existing techniques. Thus, we think it will be also be useful in the analysis of pathological conditions such as seizures, disturbed circadian rhythms etc.

The second technical advance we make through Mesmerize is the ability to provide a quantitative method to define discrete archetypical shapes from calcium traces (akin to a calcium signaling alphabet) which has been a long-standing requirement in the field, especially in the context of complex and irregular calcium activity like that observed during embryonic development or in pathological conditions (e.g. seizures). To date such archetypes were either defined in a subjective manner (as ‘puffs’, ‘blips’, etc) or by grouping peaks according to features like amplitude, width, slope etc. The second important analytical tool we have introduced is k-Shape clustering which is a novel way to perform comprehensive shape-based analysis of calcium activity traces (lines 437-464, please see Figure 6). Furthermore, Mesmerize can use k-Shape derived archetypes in combination with statistical models such as Markov Chains, with the aim to understand similarities and differences in the structure and organization of calcium dynamics across different cell types (for example please see Fig.6f-g).

Is this paper making any specific claim about biological findings (A)? Is this manuscript introducing new computational methodologies for the analysis of neuronal population in calcium imaging data (B)? Is this paper introducing a generalized and flexible platform for the analysis of calcium imaging data (C)? In each of the cases above many questions need to be answered and comparison with the state of the art must be tackled.

A) The author seems to hint at some biological findings, but they are not clearly delineated. If this is a part of the claims included in the paper it should better developed and organized. The results section should be decoupled from the methods and clear biological claims should be made, aligned with data clearly supporting the conclusions.

Our biological findings, which we believe are of high significance and novelty (as stated by reviewer 3) relate to the analysis of spontaneous calcium dynamics in neuronal and non-neuronal populations of *Ciona*. To address the concern of the reviewer we have reorganized and extended the text related to our findings in *Ciona* addressing two specific biological findings:

1. We focus on our findings showing that *Ciona* epidermal and mesenchymal cells exhibit radically different calcium fingerprints from neurons (lines 382-397) and we discuss what the role of these dynamics may be in terms of cellular functions (lines 417-421).
2. We elaborate on our interpretation of the neuronal clustering analysis. We propose that our results support previously published anatomical and molecular studies implicating: a) four cell types with a strong glutamatergic signature (PSNs, RTENs, aATENs and pATENs) in the same somatosensory circuit (lines 402-408) ; b) five classes of interneurons with high activity some of which have been postulated based on their numerous synaptic partners to act as ‘hubs’ in the larval nervous system (lines 409-417). Additionally, we have now dedicated a entire figure to our main *Ciona* findings, fully decoupled from methods (please see Figure 5).

B) In this case the authors should also relate to the state-of-the-art. It is the reviewer’s opinion that the authors are wrapping existing tools. Is the application of these tools novel for calcium imaging? For instance, the authors report "When used for hierarchical clustering, the EMD of frequency domain representations leads to better separation of disparate dynamics and an aggregation of similar dynamics (Fig 3e-g).", however, this needs to be supported with quantitative statements and related to the state of the art.

** <https://bmcbiol.biomedcentral.com/articles/10.1186/s12915-018-0606-4>*

** and others*

With regards to tools Mesmerize does four things:

1. It wraps existing tools that are very popular in the community for signal extraction and downstream analysis (e.g. Caiman modules, Suite2p for signal extraction and dimensionality reduction of neuronal activity via sklearn)
2. It repurposes tools that previously were not used in the context of calcium imaging (e.g. NuSeT)
3. It flexibly combines different tools to improve the quality of calcium imaging analysis (e.g. NuSeT with CNMF). This is not just limited to these two tools, but can simplify the process for users to combine multiple tools together.
4. It provides contemporary tools, that have previously not been used for calcium imaging analysis, such as DFT-EMD clustering and k-Shape clustering.

Tools included in packages like Caiman are very popular amongst neuroscientists and it would be a big omission not to include them in Mesmerize. However, prior to Mesmerize these were in reality only accessible to researchers with a strong computational background and what in a way was missing to popularize them to the

broader neuroscience community and beyond (E.g. developmental biologists interested in the role of calcium in a certain process) was a GUI and an easy way to organize an analysis pipeline (like Mesmerize's flowchart).

In addition, we can give an affirmative answer to the reviewer with regards to the question of whether Mesmerize encapsulates tools whose application is novel for calcium imaging. For example, while frequency-based analysis of calcium traces has been used to compare different experimental groups (e.g. <https://www.sciencedirect.com/science/article/pii/S0896627319300091> and <https://www.frontiersin.org/articles/10.3389/fncir.2013.00199/full>), it has not been used for pairwise comparison of individual traces for clustering analysis. We identified Earth Mover's Distance (EMD) as a potential solution to address this problem. Even though EMD has been used in gene expression analyses (Nabavi et al. Bioinformatics 2016) or phenotypic screen contexts (Chen et al, Nature Methods, 2020) it has not been applied in the context of calcium imaging analysis. In principle, the Earth Mover's Distance (EMD) is more appropriate for computing a distance metric between discrete Fourier transforms since the features, i.e. the frequencies, progress linearly in the feature vector and are not unrelated. For example, EMD accounts for similarities between neighboring frequency domains when computing the distance between discrete Fourier transforms (DFTs) that represent a trace. In other words, the order of elements within the feature vectors (i.e. the DFT) is important. In contrast, Euclidean distances do not account for the element order within a feature vector which makes them less suitable for measuring distances between DFTs. We illustrated this in Fig 4a-c.

k-Shape clustering which has been increasingly popular for shape-based clustering in time series analysis it has not been used in the past in the context of calcium imaging. Lastly, a deep-learning tool for cellular segmentation called NuSeT (Yang et al., PLOS Comput. Biol., 2020) which was built for segmenting nuclei to obtain cell numbers in control and pathological conditions (e.g. neurodegenerative diseases) has been repurposed in Mesmerize to segment cells expressing nuclear GCaMP6. We note that this is indicator is very popular in whole brain imaging studies across a number of species including zebrafish and *C. elegans*.

Importantly, to address the reviewer's concern we have added quantitative metrics to show how EMD-based hierarchical clustering performs in comparison to other methods. We evaluate the performance by using the agglomerative coefficient, Silhouette scores, and Davies-Bouldin scores (please Figure 4i-l, lines 366-377). The methods introduced in our manuscript outperforms traditional methods when used for the analysis of our *Ciona* dataset.

C) How to deal with multiple data formats and ontologies (especially in connection with behavioral data)? Is this platform been employed already by other users besides the developer and researchers within the same laboratory? What form of support is provided to users? How easy is to integrate new steps into the analysis pipeline? How does this framework compare with existing tools?

* <https://journals.plos.org/ploscompbiol/article?rev=1&id=10.1371/journal.pcbi.1005526>

* <https://www.frontiersin.org/articles/10.3389/fninf.2017.00044/full>

* *And more ...*

How to deal with multiple data formats and ontologies (especially in connection with behavioral data)?

We thank the reviewer for raising the interesting and indeed contemporary question of how to handle multiple data formats and ontologies. Mesmerize is a highly modular platform and thus for example it is relatively straightforward to create importers for different data formats. During the revision period one of our external users (not affiliated with our lab or institute in any way) requested an importer for Suite2P output files with the aim to import ROIs into the Mesmerize work environment, which were able to rapidly deliver.

Please see for details:

http://docs.mesmerizelab.org/en/master/user_guides/viewer/modules/suite2p_importer.html

In addition, during the same period we were endorsed by Femtonics, a widespread 2-photon microscope manufacturer in Europe, who have been using it for their work and recommending it to their users (please visit: <https://femtonics.eu/mesmerize/>). Mesmerize's modular structure made it easy to create importers for their specific data formats and we are open to creating more importers for other sources. Regarding various ontologies, users can already create their own annotations (E.g. genotypes, opto-,chemo-genetic perturbations, etc) which can be integrated into downstream analysis with no additional work. These can be added as custom categorical columns by changing the "Project Configuration" at any point along the duration of a project. We are available to discuss and aide users with setting their project configuration in our gitter community chatroom.

Please see:

http://docs.mesmerizelab.org/en/master/user_guides/project_organization/new_project/new_project.html#project-configuration

Behavioral data can be interpreted as temporal annotations, and therefore they can be added in exactly the same fashion as stimulus maps.

Please see:

http://docs.mesmerizelab.org/en/master/user_guides/viewer/modules/stimulus_mapping.html

Is this platform been employed already by other users besides the developer and researchers within the same laboratory?

The reviewer wonders whether Mesmerize is under use from other researchers outside the Chatzigeorgiou group. Over the past year we have been increasing the number of users of Mesmerize, some of the more main users can be found here:

<http://docs.mesmerizelab.org/en/master/users.html>

There are several other users as you can see from our gitter community.

As mentioned previously we have been endorsed by a major european 2-photon microscope manufacturer, which we believe will increase the number of Mesmerize users. We would also like to note that Mesmerize appears to have a large community, even before being published, compared to some of the tools listed in Table 1 when looking at the Gitter and issue tracker on Github.

https://gitter.im/mesmerize_discussion/community

<https://github.com/kushalkolar/MESmerize/issues?q=is%3Aissue+is%3Aclosed>

What form of support is provided to users?

We provide continuous support to external users through the Gitter Mesmerize community, which can be found here: https://gitter.im/mesmerize_discussion/community. We also use the issue tracker on GitHub.

We provide regular maintenance and Mesmerize releases through GitHub and the Python Package Index (PyPi):

<https://github.com/kushalkolar/MESmerize>

<https://pypi.org/project/mesmerize/>

How easy is to integrate new steps into the analysis pipeline?

Mesmerize is a modular, flexible, and easily expandable platform. New analysis tools are very easy to integrate (e.g. we have integrated new tools like 3D CNMF, NuSeT, Suite2p importer). Analysis pipelines are easy to create (using the Flowchart) and there is substantial flexibility in terms of combining different native modules and plugins modules together. An example that nicely illustrates this is our analysis pipeline for the zebrafish 3D dataset where users can flexibly choose to perform 2D CNMF per individual plane, or to use 3D CNMF alone and even combine the deep learning tool NuSeT in the same pipeline with different CNMF options.

Writing Viewer plugins for Mesmerize is simple and we have explained it in our developer documentation section. These user plugins can interact with the Viewer in the same way as native viewer modules using the Viewer Core API.

http://docs.mesmerizelab.org/en/master/developer_guide/viewer_modules.html

http://docs.mesmerizelab.org/en/master/api_reference/Viewer_data_types.html

How does this framework compare with existing tools?

* <https://journals.plos.org/ploscompbiol/article?rev=1&id=10.1371/journal.pcbi.1005526>

* <https://www.frontiersin.org/articles/10.3389/fninf.2017.00044/full>

* *And more ...*

Since both reviewers 1 and 2 requested a comparison of Mesmerize to existing tools and the state-of-the-art we provide a synoptic overview in Table1. In addition, the revised introduction examines some of the state-of-the-art tools in both calcium imaging analysis and management of imaging data (lines 40-126) highlighting the absence of a platform/framework that combines a comprehensive suite of features necessary for calcium imaging as well as project management.

We believe that Mesmerize combines in a well-balanced manner the key advantages of state-of-the-art project management tools (e.g. OMERO, Biaflows, Cytomine, OpenBIS and KNIME) with those of Calcium analysis tools (e.g. Caiman library, Suite2p, SIMA, EZCalcium, S.A. Romano). Importantly, it has the ability to interface with some of these tools like Caiman and Suite2p and we are looking forward to interface in the near future with workflow management tools for neurophysiological analysis, namely DataJoint and NWB.

Mesmerize is a modular, flexible platform designed to answer a much broader range of biological questions. Compared to some of the existing tools which feature a single pipeline used to harness neural assemblies, in Mesmerize users can build pipelines suitable for imaging neural circuits but also radically different cell types and tissues. For example in an unrelated project studying development, we have used Mesmerize to analyze calcium dynamics during tubulogenesis in *Ciona* developing embryos (Please see: <https://doi.org/10.1101/2020.10.16.342535>).

From a programmatic point of view Mesmerize is designed so that users with a moderate Python background can write new Viewer modules, Flowchart nodes and Plots. Most importantly, along with the Mesmerize API, these users-created modules can seamlessly interact with the rest of the platform. For example, if a user creates a new Viewer module for ROI extraction, ROI annotation, motion correction etc. in accordance with the Mesmerize API, the data from these modules can be smoothly used for downstream analysis and visualization – just as if the data had been processed through a standard viewer module. Thus, Mesmerize brings an entirely new type of analysis tool with a much more comprehensive suite of features and broad versatility that does not exist in the current ecosystem (Table 1). Compared to some of the tools presented in Table 1 which are written in Matlab, Mesmerize is written in Python which is being more widely adopted for analysis in neuroscience and other biological/biomedical fields and allows it to utilize cutting-edge machine learning tools such as tslearn. Most importantly, Mesmerize relies on Python libraries that are freely available, whereas Matlab based toolbox require license for several toolboxes in addition to a Matlab license, which can be a barrier for many scientists. The more open nature of the Python ecosystem also makes it easier for users to integrate Mesmerize into their own analysis and create custom modules.

This seems a very well designed application to solve the analysis problems tackled in a laboratory. The manuscript would benefit from a clear pathway to have these methods generalize across laboratories and scientific questions.

To better illustrate how these methods can be generalized we have expanded the sections on the mouse dataset analysis and added a section on 3D analysis with a zebrafish dataset. We also provide video tutorials which describe how Mesmerize's features & components can be tailored for different types of analysis. The Mesmerize documentation provides extensive assistance to help users create a "Mesmerize Project" tailored to their specific biological questions and experiments. Examples are provided on how to organize their project for various types of experiments such as stimulus/behavior responses, chemogenetics, organization of different promoters, anatomical or cell-type labels for ROIs, etc.
Please visit:

http://docs.mesmerizelab.org/en/master/user_guides/project_organization/new_project/new_project.html#biological-questions

As mentioned previously, we have also been endorsed by Femtonics, a widespread 2-photon microscope manufacturer in Europe, who have been using it for their work and recommending it to their users (please see: <https://femtonics.eu/mesmerize/>). In particular we created importers for their specific data formats thus researchers using Femtonics microscopes can effortlessly analyze their data using Mesmerize. Any imaging data format which can be expressed as a *numpy* array can be used in Mesmerize via the Viewer Core API. We also provide a video tutorial that goes through some of the basics of the Viewer Core API.

Please see:

http://docs.mesmerizelab.org/en/master/api_reference/Viewer_data_types.html

Minor comments:

* *FAIR: abbreviation not introduced*

* *TLC: abbreviation not introduced*

We thank the reviewer for pointing out these two omissions. These are abbreviations are now introduced in the text when they first appear.

* *Many figure panels are very difficult to read. The fonts and graphics are often too small. Examples: Fig 1 b, h, i, j, d ... etc.*

We provide the figures at high DPI so if the reviewer zooms in (roughly 300%) a lot of the details of the graphics will become clear.

* *“Simple data structures, outlined in the API, allow the imaging data to potentially originate from any model organism” -> not clear what does this refer to, paper should be self contained*

This sentence referred to the online documentation that can be found using the following link: <http://docs.mesmerizelab.org/en/master/index.html>. However, we understand the reviewer’s comment about the need for the paper to be self-contained, therefore we now have a supplemental pdf file that includes the entire Mesmerize documentation in approximately 200 pages. The paper has also been reorganized to better explain this idea (see section “Project Organization”).

* *Reference to Fig 1b appears before reference to Fig 1a*

We thank both reviewers 1 and 2 for pointing this out. We now mention Fig 1a before Fig 1b. In the Rich Data Annotation section (*lines 131-133*) it now reads as follows: “...zebrafish, and *Ciona intestinalis* (Fig 1a). These datasets can be visualized using the Mesmerize Viewer, which provides GUI front-ends (based on pyqtgraph) and API

interfaces for various signal extraction modules (Fig 1b).”

Reviewer #3, an expert in C intestinalis neurobiology (Remarks to the Author):

Calcium imaging is a powerful tool for analyzing cellular activity of nervous systems, embryos, and other various tissues. The authors developed a new calcium imaging analysis platform, Mesmerize, with which users can perform the whole analysis process from raw data to analyzed data presentation. The platform is flexible and expandable. The authors demonstrate usefulness of this new platform by using GCaMP6 calcium imaging data of whole larvae of Ciona intestinalis and also using mammalian brain and C. elegans data sets. The authors' work also demonstrates the power of this emerging model chordate to study development and function of the nervous systems and some other tissues. I think this work has a substantial impact for various fields of biology and life sciences, including neurobiology, physiology and developmental biology, and therefore, I support publication of this manuscript in Nature communications if the following problems are properly solved.

Most of my concerns are about organization of the manuscript and insufficient presentation of biological materials and experimental methods as summarized below.

The very long Introduction seems to contain substantial amount of detailed explanation of the Mesmerize platform. This is somewhat confusing. I think moving these parts to Results & Discussion will make the manuscript easier for readers to understand.

We thank the reviewer for these suggestions. We have better organized the manuscript to the aim of incorporating as much as possible the suggestions of all three reviewers. We now provide headings and subheadings which hopefully will clearly identify each topic discussed in the manuscript. The introduction has been revised so that it doesn't immerse the reader immediately on the various technical aspects of Mesmerize but rather it discusses the state-of-the-art in terms of calcium analysis and project management tools and we identify the main gaps in the field. Our revised results section is now divided into two main sections. The first discusses firstly how Mesmerize is largely focusing on technical aspects of the platform. The second section is focusing primarily on well separated biological examples and findings (mouse, zebrafish and Ciona) while trying to highlight how novel analytical tools (e.g. EMD and k-Shape based clustering) can be used to obtain novel insight into calcium dynamics. Of course we are open to any specific suggestions that the reviewer may have.

Some essential information on biological materials seems to be lacking. There is no description of biological procedures using the main experimental organism (Ciona), such as how the authors obtained the animals and embryos, how they transfect GCaMP6 constructs into embryos, how they reared the embryos and larvae, and what temperature was used during development and imaging analysis of embryos and larvae.

We have added two Methods subsections titled ‘Rearing conditions for adult Cionas’ and ‘Electroporation of zygotes and staging of larvae’ where we provide the relevant experimental information.

Ciona is a marine invertebrate chordate and genetically diverse between different populations. Genetically homogeneous strains have not been established. I assume that the authors used Ciona intestinalis (type B), the species whose connectome has been solved (Ryan et al., 2016, 2019). However, many other key studies, including single cell transcriptomic studies (Sharma et al. 2019; Cao et al., 2019) and the genome project (Dehal et al., 2002; Satou et al., 2019) came from the related species Ciona robusta (Ciona intestinalis type A). Therefore, it is important to precisely describe the geographic origin or population of Ciona used in the study.

The reviewer is correct. We used *Ciona intestinalis* (Type B). We have added a section in Methods titled ‘Obtaining *C. intestinalis*’. In this section we detail the precise location (including GPS coordinates) from which we obtained the adult animals used in our study.

Information on detailed developmental stages of Ciona larvae used in this study is also important because modes of larval behavior change during larval stages and thus neural circuits and neuronal functions may change during larval development (Nakagawa et al., Photochem. Photobiol. 70, 359–362, 1999; Tsuda et al., J. Exp. Biol. 206, 1409–1417, 2003; Tsuda et al., Zool. Sci. 20, 13–22, 2003). Therefore, the developmental stages of the larvae observed in this study should be described, for example, by showing time (hours) after fertilization along with incubation temperature and/or stages according to the standard developmental table of Ciona (FABA/FABA2; <https://www.bpni.bio.keio.ac.jp/chordate/faba/1.4/top.html>).

We have added the relevant information in the Methods subsection ‘Electroporation of zygotes and staging of larvae’.

Experimental procedures for the imaging of Ciona larvae should be described in some more detail. The authors just mentioned that larvae were embedded in low melting point agarose. Concentration (%) of low melting point agarose should be described. Also, information on the microscope used should be mentioned.

We thank the reviewer for highlighting this missing information. In the revised manuscript Methods section we have added a subsection titled ‘Ciona Calcium Imaging’ with substantial details regarding the Ciona calcium imaging experimental procedures.

Cell-type identification presented in this manuscript is not very convincing. In the Signal Extraction subsection of Methods, the authors mention that cells were identified with the assistance of the connectome to the best of their capability with 1-photon data. However, as shown in Table 1, most neural promoters used label many cell types. How did they identify the

cells? The connectome was obtained by using a larva developed with an egg envelop (chorion) (Ryan et al. 2016). However, reporter gene transfection into Ciona embryo is usually performed by electroporation with dechorionated eggs. It is known that dechorionation disturbs left-right asymmetry of the larval brain (Shimeld and Levin, 2006; Oonuma et al. 2016). This would make the comparison with connectome data difficult. Representative examples of identified cell type images should be presented.

We now provide a supplemental figure 3 with three examples of identified cell types. This supplemental figure comes with an extended figure legend providing some information on how we identified the cells. Briefly, we identified the cell types based on the location of the cell in relation to various landmarks (such as the neck, ocellus) and the location (rostral-caudal, dorsal-ventral). The shape of the cell was also useful. While we understand our method is not as accurate as more advanced methods, such as registration to a known larval atlas (which would be an exciting future community project for Ciona labs), the identification was done to the best of our ability by considering the features. However, we believe that the accuracy was sufficient for our biological conclusions which are at a broader level, i.e. identification of peripheral sensory neurons vs. brain-vesicle neurons and their disparate dynamics.

Minor points:

In the first paragraph of Results & Discussion, there are some mis-citations of Figure 2. Fig 2b, Fig 2c, Fig 2d, and Fig 2e may be corrected to Fig 2a, Fig 2b, Fig 2c, and Fig 2d, respectively.

We have now re-designed Figure 2 and we have checked that the we have correctly matched the citations of Figure 2 to the relevant text sections.

Reviewers' Comments:

Reviewer #1:

Remarks to the Author:

In this study, Kolar et al. present an analysis software called Mesmerize, which is a dynamically adaptable analysis platform for 2D and 3D calcium formation data. In the revised manuscript, the authors have tried to answer the questions I raised in the previous review of this manuscript. For example, they have used Mesmerize to study calcium imaging 3D data. But here, the 3D analysis is minimal, and I doubt that such analysis will be frequently used by other researchers in the field. In my opinion, the authors have not convinced me about the significant advance of using Mesmerize to study calcium signalling and its biological effects. Thus, my critique remains from the first time I reviewed this manuscript.

To support a publication in a leading international multidisciplinary scientific journal such as Nature Communications, I would want to see that Mesmerize could be used to study and answer biological questions of principal scientific value. The work presented in this manuscript does not answer any biological questions. Furthermore, the revised manuscript is still challenging to read and grasp. I firmly believe that this work could be presented better if the manuscript was written in a format often used in method journals, such as Nature Protocols.

Nature Communications aims to publish high-quality research that represents important advances of significance to specialists and researchers in biology and other fields. I do not see that this manuscript meets this criterion. Mesmerize cannot offer a significant improvement in the analysis of data from calcium imaging experiments. In fact, the analyses performed by Mesmerize can be carried out by any researcher with good knowledge of R. Mesmerize mainly uses existing tools combined in a new graphical interface. Table 1 of the manuscript lists several different platforms and pipelines that can perform somewhat similar analyses. Mesmerize is yet another platform in this list of software tools that do not offer ground-breaking enhancements.

Reviewer #3:

Remarks to the Author:

Calcium imaging is a powerful tool for analyzing cellular activity of nervous systems, embryos, and other various tissues. The authors developed a new calcium imaging analysis platform, Mesmerize, with which users can perform the whole analysis process from raw data to analyzed data presentation. The platform is flexible and expandable, and can also be used for the analysis of imaging data of neurotransmitters, voltage, and other genetically encoded indicators. The authors clearly demonstrate advantages and usefulness of the Mesmerize platform with calcium imaging data sets obtained with the mouse brain and zebrafish larvae. Furthermore, the authors conducted a comprehensive analysis of original data sets obtained with whole larvae of *Ciona intestinalis* expressing GCaMP6 in diverse types of cells under the control of various promoters. The authors' work also demonstrates the power of this emerging model chordate to study development and function of the nervous systems and other tissues.

This work should have a substantial impact for various fields of research, including neurobiology, physiology and developmental biology. I pointed out several problems in the old version of manuscript by the same group. Now, the authors addressed all the point I was concerned about and this new manuscript is substantially improved. Therefore, I support publication of this manuscript in Nature communications in the current form.

Reviewer #4:
None

Reviewer #1 (Remarks to the Author):

In this study, Kolar et al. present an analysis software called Mesmerize, which is a dynamically adaptable analysis platform for 2D and 3D calcium formation data. In the revised manuscript, the authors have tried to answer the questions I raised in the previous review of this manuscript. For example, they have used Mesmerize to study calcium imaging 3D data. But here, the 3D analysis is minimal, and I doubt that such analysis will be frequently used by other researchers in the field.

We thank the reviewer for taking the time to evaluate our revised manuscript. As the reviewer acknowledges one of the new features that we have incorporated in Mesmerize is the ability to handle 3D volumetric imaging datasets with the same annotation and analysis capabilities that we provide for 2D datasets. We have used a 3D zebrafish dataset from a high-impact publication (Haesemeyer et al., Neuron 2018) to validate this new functionality technically and biologically.

In addition, Mesmerize now provides a solution to the long-standing problem of relatively poor performance of the standard ROI/signal extraction packages when performing 2-photon volumetric imaging (Keller & Ahrens, Neuron, 2015). Our integration of a deep-learning tool for cellular segmentation called NuSeT (Yang et al., PLOS Comput. Biol., 2020) with CNMF markedly improved ROI identification and signal extraction (please see Figure3b). Therefore, we are of the opinion that neuroscientists and biomedical researchers who wish to analyse their 3D calcium imaging data will use Mesmerize because it provides a user-friendly, technically, and biologically validated tool to handle and analyse 3D calcium imaging datasets. Researchers in the field are even more likely to use Mesmerize because our platform comes with improved performance relative to the state-of-the-art when it comes to ROI identification and signal extraction in volumetric datasets.

Importantly, as we mentioned in our previous response to reviewers, the endorsement of Mesmerize by Femtonics, a widespread 2-photon microscope manufacturer in Europe with a strong focus on 3D imaging applications, who have been using it for their work and recommending it to their users makes us hopeful that Mesmerize will be used frequently by other researchers in the field.

In my opinion, the authors have not convinced me about the significant advance of using Mesmerize to study calcium signalling and its biological effects. Thus, my critique remains from the first time I reviewed this manuscript.

We respectfully disagree with the reviewer. In our manuscript we include two analytical approaches for which we demonstrate their novelty and usefulness to study calcium signalling and its biological effects.

Firstly, a novelty of our work is the establishment of the Discrete Fourier Transform (DFT) – Earth Mover's Distance (EMD) clustering approach, which we show it can outperform more commonly used methods for clustering calcium signaling activity as we demonstrate in our manuscript (Fig. 4i-l). We use this approach to quantify and obtain novel biological insight into the diversity of calcium dynamics from large numbers of neuronal and non-neuronal cell types in the *Ciona intestinalis* tadpole. As we mentioned in our previous response to reviewers the DFT – EMD clustering approach is suitable for data that are not temporally aligned it has significant potential for the analysis of spontaneous activity during circadian cycles, development (of both neuronal & non neuronal tissues)

and pathological conditions using as psychiatric disease models. This suggests that it will be of particular use in biomedical studies.

The second technical advance we make through Mesmerize is the application of k-Shape clustering to calcium imaging data. This machine learning approach provides a quantitative method to define discrete archetypical shapes from calcium traces (akin to a calcium signaling alphabet) which has been a long-standing requirement in the field. We expect that our approach will replace qualitative descriptions of shapes such as 'blips', 'puffs' which are extensively used to date. Furthermore, using our novel *Ciona intestinalis* dataset biologists can use k-Shape derived archetypes in combination with statistical models, such as Markov Chains, to understand similarities and differences in the structure and organization of calcium dynamics across different cell types as illustrated in our manuscript (Fig.6f-g).

To support a publication in a leading international multidisciplinary scientific journal such as Nature Communications, I would want to see that Mesmerize could be used to study and answer biological questions of principal scientific value. The work presented in this manuscript does not answer any biological questions.

We respect the opinion of the reviewer; however, we have a different view and we believe that our work "should have a substantial impact for various fields of research, including neurobiology, physiology and developmental biology" as stated by reviewer #3.

From an evolutionary and neurobiological perspective, we believe that the biological work presented in our manuscript is of high significance and novelty within the *Ciona* and protochordate field. *Ciona* is only the 2nd organism after *C. elegans* with a fully mapped connectome (Ryan et al., eLife, 2015; Ryan et al., Current Biology, 2016; Ryan et al., J Comp Neurol, 2017) and benefits from recently completed single cell molecular blueprint of the entire nervous system (Sharma et al, Dev. Biol., 2019; Cao et al, Nature, 2019). Our work complements these studies by providing the first neuronal activity blueprint of the *Ciona* tadpole nervous system and illustrates the importance of this organism as an emerging model to study the development and function of the chordate nervous system and other tissues as indicated also by reviewer #3.

Furthermore, the revised manuscript is still challenging to read and grasp. I firmly believe that this work could be presented better if the manuscript was written in a format often used in method journals, such as Nature Protocols.

We have made a substantial effort to help the broader readership understand the key features of Mesmerize as a calcium imaging analysis platform and how it can be used to analyse a diversity of nervous systems (mouse, zebrafish, *C. intestinalis*) and cell types. In addition, we have placed particular emphasis in detailing our original findings regarding calcium dynamics in the *Ciona* tadpole nervous system. Therefore, we don't agree that our manuscript should be written in a format like that of Nature Protocols.

Nature Communications aims to publish high-quality research that represents important advances of significance to specialists and researchers in biology and other fields. I do not see that this manuscript meets this criterion. Mesmerize cannot offer a significant improvement in the analysis of data from

calcium imaging experiments. In fact, the analyses performed by Mesmerize can be carried out by any researcher with good knowledge of R.

We believe that in earlier sections of our response to the reviewer we have explained the important advances of significance to specialists and researchers in biology that Mesmerize is offering. Importantly, the presence of several different platforms and pipelines that we compare Mesmerize to, suggests that calcium imaging analysis is a complex task that undoubtedly requires from researchers much more than just a good knowledge of R.

While our platform is used from calcium imaging analysis specialists and programming experts, we would like to stress that Mesmerize has been designed for a general audience that may not have prior programming experience. Especially for this broad audience of biology researchers that needs to combine reproducible calcium imaging analysis with FAIR dataset generation and interactive visualizations, we believe that Mesmerize will be a game changer.

Mesmerize mainly uses existing tools combined in a new graphical interface. Table 1 of the manuscript lists several different platforms and pipelines that can perform somewhat similar analyses. Mesmerize is yet another platform in this list of software tools that do not offer ground-breaking enhancements.

We would like to state that we think that most of the software tools listed in Table 1 have contributed to the ever-growing ecosystem of calcium imaging analysis tools and platforms and we have found their work to be useful and inspiring. With regards to tools and contribution of Mesmerize we would like to reiterate four things we have mentioned in our previous response to reviewers:

1. Mesmerize wraps existing tools that are very popular in the community for signal extraction and downstream analysis (e.g. Caiman modules, Suite2p for signal extraction and dimensionality reduction of neuronal activity via sklearn)
2. It repurposes tools that previously were not used in the context of calcium imaging (e.g. NuSeT)
3. It flexibly combines different tools to improve the quality of calcium imaging analysis (e.g. NuSeT with CNMF). This is not just limited to these two tools but can simplify the process for users to combine multiple tools together. This is possible because Mesmerize is a flexible platform with a large API that allows tools to be both integrated and combined.
4. It provides contemporary tools, that have previously not been used for calcium imaging analysis, such as DFT-EMD clustering and k-Shape clustering.

The feedback that we have received from external users of Mesmerize is that this synthetic approach that the Mesmerize platform follows is extremely useful and we plan to build on it.

Reviewer #3 (Remarks to the Author):

Calcium imaging is a powerful tool for analyzing cellular activity of nervous systems, embryos, and

*other various tissues. The authors developed a new calcium imaging analysis platform, Mesmerize, with which users can perform the whole analysis process from raw data to analyzed data presentation. The platform is flexible and expandable, and can also be used for the analysis of imaging data of neurotransmitters, voltage, and other genetically encoded indicators. The authors clearly demonstrate advantages and usefulness of the Mesmerize platform with calcium imaging data sets obtained with the mouse brain and zebrafish larvae. Furthermore, the authors conducted a comprehensive analysis of original data sets obtained with whole larvae of *Ciona intestinalis* expressing GCaMP6 in diverse types of cells under the control of various promoters. The authors' work also demonstrates the power of this emerging model chordate to study development and function of the nervous systems and other tissues.*

This work should have a substantial impact for various fields of research, including neurobiology, physiology and developmental biology. I pointed out several problems in the old version of manuscript by the same group. Now, the authors addressed all the point I was concerned about and this new manuscript is substantially improved. Therefore, I support publication of this manuscript in Nature communications in the current form.

We would like to thank the reviewer for their time and thoughtful suggestions and feedback. We agree with the reviewer that *Ciona* is an emerging model chordate to study the development and function of the nervous system and other tissues/organs and we plan to harness the strengths of Mesmerize to serve the community technically and biologically in the best way possible.